# Immunodeficient mice are better for modeling the transfusion of human blood components than wild-type mice

Sophia A. Blessinger[1], Johnson Q. Tran[1], Rachael P. Jackman[1,2], Renata Gilfanova[1], Jacqueline Rittenhouse[1], Alan G. Gutierrez[1], John W. Heitman[1], Kelsey Hazegh[3], Tamir Kanias[3,4], Marcus O. Muench[1,2]*

1 Vitalant Research Institute, San Francisco, CA, United States of America, 2 Department of Laboratory Medicine, University of California, San Francisco, CA, United States of America, 3 Vitalant Research Institute, Denver, CO, United States of America, 4 Department of Pathology, University of Colorado Denver Anschutz Medical Campus, Aurora, CO, United States of America

* mmuench@vitalant.org

**Data Availability Statement:** All relevant data are within the manuscript and its Supporting Information files.

## Abstract

Animal models are vital to the study of transfusion and development of new blood products. Post-transfusion recovery of human blood components can be studied in mice, however, there is a need to identify strains that can best tolerate xenogeneic transfusions, as well as to optimize such protocols. Specifically, the importance of using immunodeficient mice, such as NOD.Cg-*Prkdc^scid Il2rg^tm1Wjl*/SzJ (NSG) mice, to study human transfusion has been questioned. In this study, strains of wild-type and NSG mice were compared as hosts for human transfusions with outcomes quantified by flow cytometric analyses of CD235a+ erythrocytes, CD45+ leukocytes, and CD41+CD42b+ platelets. Complete blood counts were evaluated as well as serum cytokines by multiplexing methods. Circulating human blood cells were maintained better in NSG than in wild-type mice. Lethargy and hemoglobinuria were observed in the first hours in wild-type mice along with increased pro-inflammatory cytokines/chemokines such as monocyte chemoattractant protein-1, tumor necrosis factor α, keratinocyte-derived chemokine (KC or CXCL1), and interleukin-6, whereas NSG mice were less severely affected. Whole blood transfusion resulted in rapid sequestration and then release of human cells back into the circulation within several hours. This rebound effect diminished when only erythrocytes were transfused. Nonetheless, human erythrocytes were found in excess of mouse erythrocytes in the liver and lungs and had a shorter half-life in circulation. Variables affecting the outcomes of transfused erythrocytes were cell dose and mouse weight; recipient sex did not affect outcomes. The sensitivity and utility of this xenogeneic model were shown by measuring the effects of erythrocyte damage due to exposure to the oxidizer diamide on post-transfusion recovery. Overall, immunodeficient mice are superior models for xenotransfusion as they maintain improved post-transfusion recovery with negligible immune-associated side effects.

**Funding:** This work was supported by the National Institutes of Health R01 HL134653 (T.K.), R01 HL133024 (R.P.J.), and R01 HL144501 (R.P.J.). R. G. was supported by a Bridges to Stem Cell Research and Therapy Award EDUC2-08400 from the California Institute of Regenerative Medicine. The funders had no role in study design, data collection and analysis, decision to publish, or preparation of the manuscript. The content is solely the responsibility of the authors and does not necessarily represent the official views of the National Heart, Lung and Blood Institute, the National Institutes of Health, the California Institute for Regenerative Medicine or any other agency of the State of California.

**Competing interests:** The authors have declared that no competing interests exist.

# Introduction

Mouse models of transfusion play an important role in testing novel therapeutics, particularly to study aspects of transfusion medicine under controlled conditions that are not feasible in a clinical setting. Many such studies have employed mouse-to-mouse transfusion protocols that benefit from the wide array of available inbred strains. Other studies have employed transgenic mice expressing human proteins on mouse blood cells to study common complications observed in transfusion medicine such as antibody-mediated rejection of red blood cells (RBCs) [1]. Xenogeneic transfusion studies have also been performed between various animal species, including human cells into mice [2]. Such studies generally have employed immuno-deficient mice lacking various cellular elements of the adaptive immune system, such as severe combined immunodeficiency (SCID) mice.

Immunodeficient mouse strains have advanced over the last decades such that mice with multiple genetic mutations favorable to the engraftment of xenogeneic cells have been developed, including the NOD.Cg-*Prkdc^scid Il2rg^tm1Wjl*/SzJ (NSG) mouse [3]. These mice lack all T-, B-, and NK-cells and also exhibit defects in macrophages and dendritic cell function. Additionally, they have severely compromised complement component 5 (C5) expression due to a deletion in the Hc gene [4]. Such mice allow extensive humanization of the hematopoietic system [3, 5], although circulating blood cells remain primarily of mouse origin [6]. This latter observation underscores the physiological barriers to fully humanizing the hematopoietic system of mice.

Comparisons between SCID and wild-type mice found that transfused human platelets cleared 5 hours slower in SCID mice than in wild-type mice [7]. These findings support the assumption that xenogeneic tissues survive better in immunocompromised mice. However, the xenogeneic platelets were still cleared within a day in the immunodeficient mice likely due to engulfment by phagocytes; macrophages, while partially defective, still exist in these mice [8]. Studies of transfused human erythrocytes have been performed using immunodeficient and wild-type mice despite the rapid clearance of the human RBCs in mice [9–12]. Thus, given the importance of the innate immune system in the clearance of human cells from the circulation of mice, this study sought to compare xenogeneic transfusion into wild-type versus immunodeficient mice to validate the use of immunodeficient mice in the study of human blood products and to better understand the barriers to full humanization of the hematopoietic system of mice.

# Materials and methods

## Ethics statements

Blood was obtained from healthy volunteers with written informed consent under approval of the Institutional Review Board at the University of California San Francisco (IRB# 11–06262). Two 8.5 ml tubes of blood were collected for each experiment in BD Vacutainers ACD solution A (REF 364606).

Alternatively, leukocyte-reduced packed red blood cell (LR-pRBC) units from volunteer blood-donors were obtained from the blood bank (Vitalant). Units were used without identifying information to assure anonymity of the donors. Thus, these samples did not meet the definition of human subjects research as the blood was not collected specifically for experimental purposes and the donor was anonymous. These units were suspended in adenine saline, held at 4˚C, and transfused prior to expiration.

Animal research was performed with approval and oversight of the Institutional Animal Care and Use Committee at Covance Laboratories, Inc. under Animal Welfare Assurance A3367-01 and protocol number IAC 2112 / ANS 2411. Animal husbandry was carried out

according to the recommendations in the Guide for the Care and Use of Laboratory Animals of the National Institutes of Health.

All mice originated from the Jackson Laboratory: B6(C)-*H2-Ab1*$^{bm12}$/KhEgJ (bm12), C57BL/6J (B6), BALB/cJ (BALB/c), FVB/NJ (FVBN), NSG, NOD.Cg-*Prkdc*$^{scid}$ *Il2rg*$^{tm1Wj}$*l* Tg (CMV-IL3,CSF2,KITLG)1Eav/MloySzJ (NSG-3GMS or NSG-3GS), NOD.Cg-*Prkdc*$^{scid}$ *Il2rg*$^{tm1Wjl}$ Tg(PGK1-KITLG*220)441Daw/SzJ (hKL-NSG or hSCF-Tg-NSG), CByJ.B6-Tg (UBC-GFP)30Scha/J (UBI-GFP), and NOD.Cg-*Prkdc*$^{scid}$ *Il2rg*$^{tm1Wjl}$ Tg(CAG-EGFP)1Osb/SzJ (NSG-EGFP). Immunodeficient and bm12 mice were bred and housed at our institute in a vivarium free from >40 murine pathogens as determined through biannual nucleic acid testing (Mouse Surveillance Plus PRIA; Charles River) of sentinel mice housed with bedding mixed from cages throughout the vivarium. Mice were maintained in sterile, disposable microisolater-cages containing irradiated corn-cob bedding (Innovive Inc.) and fed a sterile, irradiated diet of Teklad Global 19% protein diet (Envigo) with free access to sterile-filtered, acidified water (Innovive Inc.). Environmental enrichment was provided by autoclaved cotton Nestlets (Ancare Corp.) and GLP-certified Bio-Huts (Bio-Serv) added to cages during fortnightly cage-changes.

Procedures performed on mice involved only brief restraint with the exception of exsanguination, which was done under a deep plane of inhalation anesthesia via orbital enucleation followed by cervical dislocation. Otherwise, mice were euthanized at the end of the experiments according to the recommendations of the American Veterinary Medical Association.

### Preparation of donor cells

Donor-cell sources and methods of preparation used in the different experiments are listed in Table 1. Leukocyte-enriched whole blood (LE-WB) cells were prepared from buffy coats by centrifugation for 30 minutes at 600 x *g*, and included the lower fraction of platelet-rich plasma and upper fraction of erythrocytes. These cells were mixed together with 14% citrate phosphate dextrose adenine-1 (CPDA-1).

Leukocyte-reduced whole blood (LR-WB) containing primarily washed RBCs were also prepared in the laboratory from small whole blood samples, collected and processed as

**Table 1. Donor blood products and number of human cells transfused in each experiment.**

| Figure | Blood Product | Erythrocytes* | Leukocytes* | Platelets* | Total Cells‡ |
|---|---|---|---|---|---|
| Fig 1B | LE-WB | N.D. | N.D. | N.D. | $2.0 \times 10^8$ |
| Fig 1C | LE-WB‡ | N.D.‡ | N.D. | N.D. | $4.2 \times 10^8$ |
| Fig 1D | LE-WB | N.D. | N.D. | N.D. | $2.1 \times 10^8$ |
| Fig 1E | LE-WB | $4.78 \times 10^8$ | $2.61 \times 10^6$ | $8.60 \times 10^7$ | N.D. |
| Figs 2 and 4 | LE-WB | $6.10 \times 10^8$ | $1.86 \times 10^6$ | $5.54 \times 10^7$ | N.D. |
| Fig 3A and 3B | LE-WB | $5.58 \times 10^8$ | $1.74 \times 10^6$ | $6.12 \times 10^7$ | N.D. |
| Fig 3C and 3D | Washed LE-WB§ | $8.82 \times 10^8$ | $1.53 \times 10^6$ | $1.00 \times 10^7$ | N.D. |
| Fig 3E and 3F | LR-pRBC | $7.58 \times 10^8$ | trace | trace | N.D. |
| Fig 5A–PBS | Washed LE-WB§ | $6.28 \times 10^8$ | $7.26 \times 10^5$ | $1.00 \times 10^6$ | N.D. |
| Fig 5A–Plasma | Washed LE-WB§ | $8.22 \times 10^8$ | $9.66 \times 10^5$ | $0.80 \times 10^6$ | N.D. |
| Fig 5B and 5C | Unwashed LR-pRBC‡ | $1.19 \times 10^9$ | trace | trace | N.D. |
| Fig 5B and 5C | Washed LR-pRBC | $1.25 \times 10^9$ | trace | trace | N.D. |
| Fig 6A and 6B | High Dose 7-day LR-pRBC | $1.08 \times 10^9$ | trace | trace | N.D. |
| Fig 6A and 6B | Low Dose 7-day LR-pRBC | $5.54 \times 10^8$ | trace | trace | N.D. |
| Fig 7 (Donor 1) | LR-pRBC | $1.08 \times 10^9$ | trace | trace | N.D. |
| Fig 7 (Donor 2) | LR-pRBC | $1.07 \times 10^9$ | trace | trace | N.D. |
| Fig 7 (Donor 3) | LR-pRBC | $1.19 \times 10^9$ | trace | trace | N.D. |

*(Continued)*

**Table 1.** (Continued)

| Figure | Blood Product | Erythrocytes* | Leukocytes* | Platelets* | Total Cells‡ |
|--------|--------------|--------------|-------------|-----------|--------------|
| Fig 8A | 7-day LR-pRBC | $1.08 \times 10^9$ | trace | trace | N.D. |
| Fig 8A and 8B | 21-day LR-pRBC | $1.07 \times 10^9$ | trace | trace | N.D. |
| Fig 8B | Irradiated 21-day LR-pRBC | $1.09 \times 10^9$ | trace | trace | N.D. |
| Fig 8B | 2X Irradiated LR-pRBC | $1.07 \times 10^9$ | trace | trace | N.D. |
| Fig 8C | LR-pRBC | $1.20 \times 10^9$ | trace | trace | N.D. |
| Fig 8C | LR-pRBC–Diamide | $1.09 \times 10^9$ | trace | trace | N.D. |
| Fig 9A, 9B and 9D | LR-pRBC | $8.58 \times 10^8$ | trace | trace | N.D. |
| Figs 9C and 11D–11F | LR-pRBCs+EGFP⁺ LR-WB** | $1.58 \times 10^9$ | $2.79 \times 10^5$ | $2.86 \times 10^7$ | N.D. |
| | (Human LR-pRBCs Alone) | $1.15 \times 10^9$ | $0.40 \times 10^5$ | $1.60 \times 10^6$ | |
| | (Mouse LR-WB Alone) | $2.31 \times 10^8$ | $1.19 \times 10^5$ | $1.23 \times 10^7$ | |
| Fig 11A–11C | LR-pRBCs+EGFP⁺ LR-WB** | $1.09 \times 10^9$ | $0.17 \times 10^5$ | $1.12 \times 10^7$ | N.D. |
| | (Human LR-pRBCs Alone) | $8.58 \times 10^8$ | $0.20 \times 10^5$ | $0.40 \times 10^6$ | |
| | (Mouse LR-WB Alone) | $2.11 \times 10^8$ | $0.19 \times 10^5$ | $7.34 \times 10^6$ | |
| Fig 12 | LR-pRBCs+EGFP⁺ LR-WB** | $1.23 \times 10^9$ | $0.88 \times 10^5$ | $1.44 \times 10^7$ | N.D. |
| | (Human LR-pRBCs Alone) | $1.05 \times 10^9$ | $0.40 \times 10^5$ | $1.60 \times 10^6$ | |
| | (Mouse LR-WB Alone) | $2.10 \times 10^8$ | $1.19 \times 10^5$ | $1.23 \times 10^7$ | |

*Cells were counted using a Heska Element HT5 hematology analyzer.

†Cells were counted using a hemocytometer and counts include leukocytes and erythrocytes.

‡Abbreviations: LE-WB, leukocyte-enriched whole blood; N.D., not determined; LR-pRBC, leukoreduced packed red blood cells; LR-WB, leukocyte-reduced whole blood

§Cells were washed prior to transfusion. Note the lower numbers of platelets.

¶LR-WB was prepared from whole blood, stored overnight at 4°C in 14% CPDA-1, by centrifugation and removal of the buffy coat and upper portion of red cells.

**Cell counts for the final mixture of 10:1 human LR-pRBC: mouse eGFP LR-pRBC are shown as well as counts made of the human and mouse cells prior to mixing. Note, low levels of leukocytes and platelets can result in less accurate measurements.

described above, by removal of the buffy coat and underlying red cells, followed by washing the lower fraction of RBCs with 50 ml of phosphate buffer saline (PBS) thrice.

As indicated in Table 1, some donor cells were washed twice with PBS and suspended in 2.5 ml PBS with 14% CPDA-1. For direct comparison of the effects of plasma removal, the washed blood cells were diluted 1:1 with either PBS with 14% CPDA-1 or autologous plasma with 14% CPDA-1 that was filtered using a 0.33μ low-protein binding filter.

Erythrocyte injury was induced by two methods to test the clearance of injured compared to uninjured cells. LR-pRBCs were irradiated using a standard blood bank γ-irradiator at a single dose of 3000cGy. A second group was irradiated twice for a total dose of 6000cGy. Another method was to treat LR-pRBCs with 2mM diamide (N,N,N',N'-Tetramethylazodicarboxamide; Sigma Aldrich) for 2 hours at 37°C prior to injection [13].

EGFP⁺ mouse erythrocytes were added to human donor cells in some experiments to track the relative fate of mouse and human cells [14]. These cells were obtained from UBI GFP mice, which are on a BALBc background and express EGFP in all hematopoietic cells. Erythrocytes were isolated by the same method as described for LR-WB human cells.

## Transfusions

Transfusions were performed by gently warming the mice using heat lamps, under constant observation, and transfusing by tail-vein injection in a 200 μl volume using insulin syringes with a 28g needle. For experiments using mouse tracer cells, mice were injected with 200 μl

human LR-pRBC plus 20 μl of washed LR-WB obtained from UBI GFP mice. Transfusion involved brief restraint, after which mice were returned to their cages.

## Flow cytometric analysis of blood samples

Blood (5–10 μl) was collected from mouse tail tips after excision and immediately mixed in PBS with 5% mouse serum, 0.01% $NaN_3$, and 14% CPDA-1. Anti-human antibodies were purchased from BioLegend unless otherwise stated. Human erythrocytes were identified using CD235ab-phycoerythrin (PE) (clone HIR2; e-Bioscience, ThermoFisher Scientific), CD235a-PE (clone HI264), or CD235ab-allophycocyanin (APC) (clone HIR2). Platelets were identified by co-expression of CD41-fluorescein isothiocyanin (FITC) (clone HIP8) and CD42b-PE or -APC (clone HIP1). Human leukocytes were stained with CD45-phycoerythrin-cyanin-7 (PC7) (clone 2D1). Subsets of light-density $CD45^+$ cells, isolated from spleens [15], were stained with CD3 (clone HIT3a), CD14 (clone HCD14), and CD33 (clone WM53). Cells were stained for ≥30 minutes with saturating concentrations of antibody based on recommendations of the supplier.

## Serum cytokine measurements

For cytokine analysis, serum was collected from mice before and 1.5 hours after transfusion of human blood, and cytokines were measured using the MILLIPLEX MAP mouse cytokine/chemokine 32-plex magnetic bead panel according to Millipore Sigma's instructions. An initial blood sample of ~100 μl was taken from the tail tip prior to transfusion and post-transfusion samples were collected by exsanguination [16].

## Complete blood counts, blood volume measurements, and tissue cell counts

Complete blood counts were performed on a Heska Element HT5 hematology analyzer using a 15 μl sample of EDTA whole blood. Blood volume was estimated based on the dilution of a known quantity of transfused $EGFP^+$ mouse erythrocytes throughout the circulation using flow cytometric and CBC analyses of the donor cells and the blood of recipients 5 minutes after transfusion. A second approach to estimating blood volume was based on the weight of the mice and a 68.9 μl / g average estimate reported by Sluiter et al. [17] for measurements made on several mouse strains and sex combinations. The content of human cells in lungs, liver, and spleen was based on hemocytometer cell counts of live cells, based on trypan blue staining, and flow cytometric measurements of human erythrocyte frequencies among total live cells based on propidium iodide staining of dead cells. Tissues were processed by passage through sieves to produce single cell suspensions.

## Data presentation and statistical analysis

Flow cytometry data were analyzed using FlowJo software (FlowJo, Inc.) to determine the frequencies of human blood cells among all single-live cells. Normalized data were generated to account for differences in recipient mice by establishing the first blood measurement for each mouse, at 5–20 minutes post-transfusion, as 100% and reporting the additional measurements relative to this value. Alternatively, data are presented simply as the raw measured frequencies. When labeled mouse erythrocytes were co-injected with human LR-pRBCs, data are also presented as a ratio of human to mouse cells.

Statistical analysis and charting were performed using Aabel NG software (Gigawiz Ltd. Co.) and Prism (GraphPad Software, Inc.). The non-parametric Mann-Whitney U test was

used to compare the frequencies of cell populations and engraftment. Comparisons made on the first blood measurement used the raw frequency measurements made at the time point. All subsequent comparisons used normalized data. Cytokine and chemokine expression were analyzed using 2-way ANOVA using Sidak's multiple comparisons test to determine the significance of differences between pre- and post-transfusion values. P-values of $\leq 0.05$ were considered significant. Linear and logarithmic regression analyses were performed using Aabel NG software, R-values are presented for linear regressions.

Frequency data at different time points from multiple mice are presented as the mean ± standard error. Time values are shown on the x-axis in a non-linear fashion to allow better visualization of early time-points. Some data are presented using notched box and whisker plots in which the notch corresponds to the median and the box extends from the 25th to the 75th percentiles. Individual data points, representing individual mice, are indicated and whiskers extend to the extreme data points. Means are also indicated using a flattened diamond symbol. The number and sex of mice transfused in each experiment are indicated in figure legends and/or in the text.

## Results

### Survival of human blood cells is longer in immunodeficient than wild-type mice

Peripheral blood cell chimerism was analyzed in wild-type mice and immunodeficient NSG mice transfused with human blood cells. The bm12 strain of mice was used in initial experiments. This fecund wild-type strain shares a genetic background with B6 mice, differing in 3 amino acids in the class II I-A$^b$ major histocompatibility antigen. Mice were transfused with LE-WB so that the kinetics of clearance of multiple human blood cell lineages could be studied using flow cytometry (Fig 1A).

In an initial experiment, chimerism was analyzed 19 hours after transfusion revealing that human cells survived longer in an immunodeficient host (Fig 1B). This observation was substantiated in two more experiments in which the kinetics of human-cell clearance were analyzed over 2–4 hours. Within the first 2 hours, wild-type mice exhibited a rapid decrease in the frequency of human CD235a$^+$ erythrocytes while normalized erythrocyte frequencies increased in NSG mice throughout the recorded time period (Fig 1C and 1D). We refer to the oft observed increase in chimerism during the first few hours after transfusion, relative to the initial measurement, as the rebound effect. The outcome of transfused human CD45$^+$ leukocytes differed the most between experiments showing some of the highest rebound effects. In the second experiment, a nearly 10-fold increase in leukocytes was observed between the 5 and 30 minute time points in NSG mice, with a rapid clearance occurring over the next 4 hours (Fig 1C). In the third experiment, bm12 mice retained a higher percentage of human leukocytes than NSG mice throughout the 2 hour period (Fig 1D). Normalized frequencies of human CD41$^+$CD42b$^+$ platelets were maintained better in NSG mice, displaying a rebound effect–similar to that seen with erythrocytes–at the 2–4 hour time points (Fig 1C and 1D).

Additional strains of wild-type mice were compared to NSG mice over a 20-hour period after transfusion (Fig 1E). NSG mice maintained higher frequencies of human blood components throughout the experiment. The differences between NSG mice and either B6 or BALB/c mice were significant for all cell types at the 120 minute time point, whereas the differences between NSG and FVBN mice were significant only for platelets. Notably, among the wild-type strains, FVBN mice tended to have the highest human cell frequencies. After 20 hours, the frequencies of all human cells were significantly higher in NSG mice than in any of the wild-type mice.

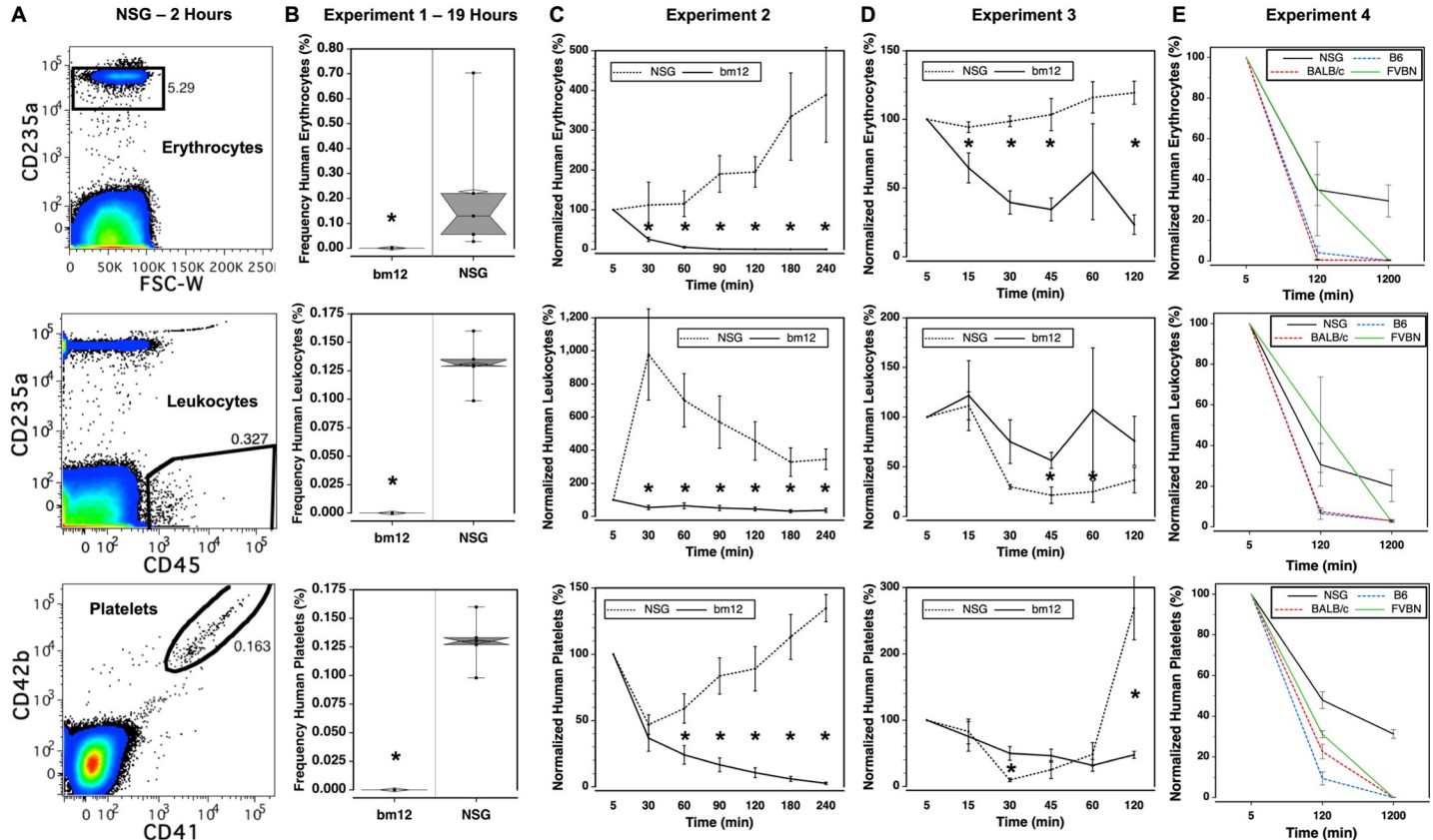

**Fig 1. Kinetics of human cell clearance in wild-type and immunodeficient mice.** (A) Flow cytometry was used to measure the frequency of human erythrocytes ($CD235a^+$), leukocytes ($CD45^+$), and platelets ($CD41^+CD42b^+$) using the representative gating strategy shown for cells recovered from an NSG mouse 2 hours after transfusion. The frequencies of each of these cell types in the peripheral blood of NSG and bm12 mice were analyzed in three experiments after 19 hours, n = 5 mice / group (B); over the course of 4 hours, n = 5–6 mice / group (C); and over the course of 2 hours, n = 5 mice / group (D). Asterisks indicated P ≤ 0.05 for each indicated time. The clearance of human cells was further compared in four mouse strains over the course of 20 hours, n = 5 mice / group, in a fourth experiment (E). Statistical analyses and animal weights for the fourth experiment are shown in S1-S4 Tables in S1 File.

While performing these experiments it was observed that wild-type mice often appeared lethargic within the first couple of hours after transfusion when compared to NSG mice. Indeed, one bm12 mouse died while in its cage about 1 hour after transfusion suggesting a deadly, acute response to the xenogeneic cells. It was also noted in these experiments that the urine of wild-type mice was strongly discolored indicating the presence of free hemoglobin from hemolysis, whereas the urine of NSG mice was generally less discolored. Examples of these observations are shown in Fig 2.

## Human cytokine expression by immunodeficient mice has no effect on the longevity of transfused human leukocytes

The NSG mouse strains NSG-3GMS and hKL-NSG express human cytokine transgenes that, due to species specificities, do not affect murine cells [18, 19]. Long-term hematopoietic engraftment of these strains shows improved multilineage engraftment and we sought to determine if short-term engraftment, particularly of leukocytes responsive to the transgenic human cytokines, from a transfusion may also be affected by the presence of human cytokines [6]. Two experiments were performed analyzing blood chimerism over time and the content of human cells in the spleen one day following transfusion of LE-WB (Fig 3A–3D). Although

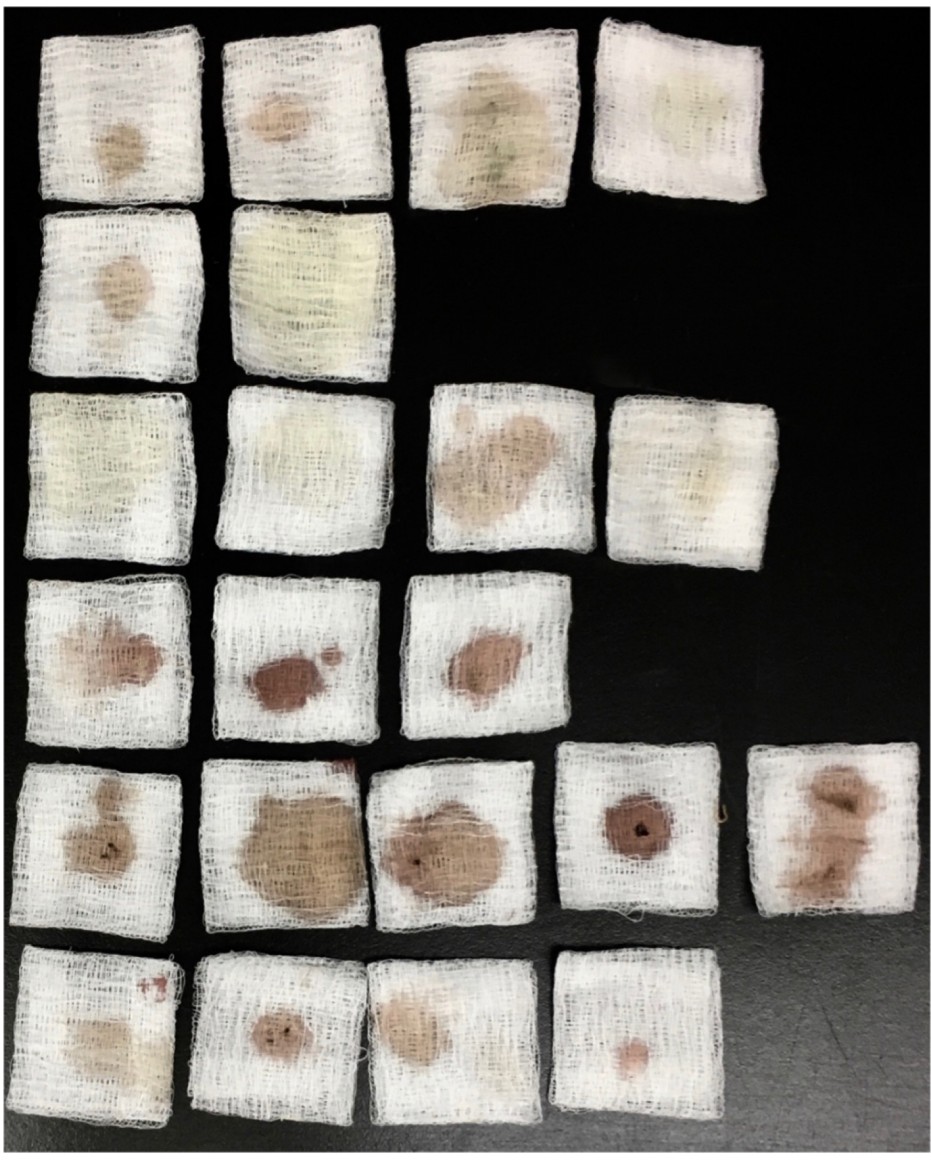

**Fig 2. Urine spots from mice after human blood transfusion.** The indicated strains of mice were transfused with LE-WB and urine was collected at the time of sacrifice, 1.5 hours after transfusion. Each gauze square represents urine from a single animal.

significant differences among blood cell counts were observed at various time points, there was no consistent trend favoring one strain over the other in the two experiments. Indeed, leukocytes were expected to be most affected by the expression of human cytokines based on long-term hematopoietic engraftment studies, yet leukocytes frequencies in the blood (Fig 3A and 3C) and spleen (Fig 3B and 3D) were mostly unaffected by the host mouse strain.

Although the transfusion of LE-WB allows for engraftment analysis of multiple lineages of human blood cells, it became apparent during the course of these studies that some of the variability in the kinetics of blood cell clearance, in particular of leukocytes, was likely due to complex interactions among the human cells, host cells, and tissues leading to the rebound effects observed. Therefore, we transfusion a single blood component, LR-pRBCs, instead of whole blood to test if the clearance of a single cell type lessened the rebound effect. No rebound effect was observed with transfusion of LR-pRBC into the three strains of immunodeficient mice

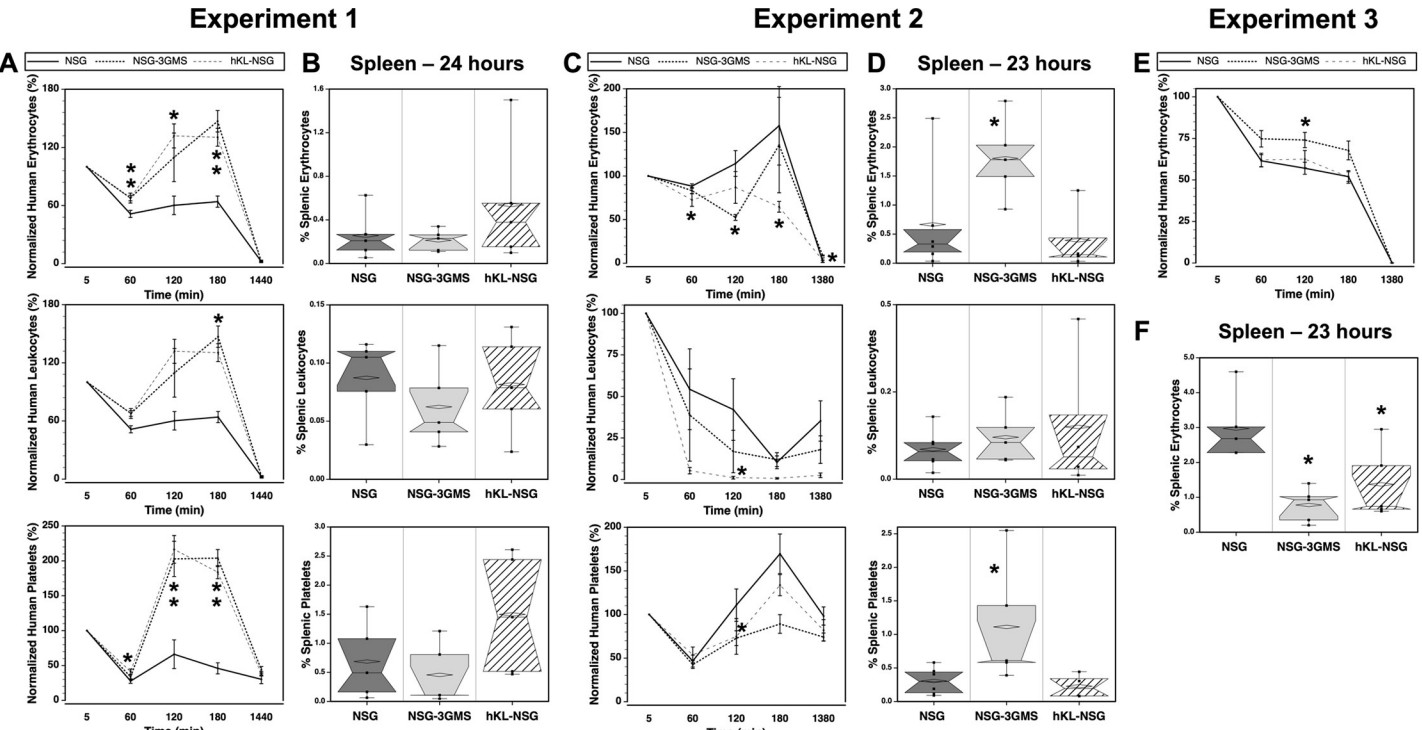

**Fig 3. Human cell clearance in strains of immunodeficient mice.** (**A, C, E**) Frequencies of human cells in the blood were compared in three immunodeficient mouse strains in three experiments. In the first two experiments mice, n = 4–6 for each strain, were transfused with LE-WB and in the third experiment mice, n = 5 for each strain, received LR-pRBC. (**B, D, F**) Human cells were analyzed in the spleen 23–24 hours after transfusion. Asterisks indicated P ≤ 0.05 compared to NSG mice.

(Fig 3E and 3F). A few significant differences were observed among the measurements made, but no consistent patterns emerged among the three experiments. This suggests that the fate of transfused human cells did not differ in these three strains of immunodeficient mice.

## Increased cytokine production in wild-type mice compared to NSG mice after human cell transfusion

Cytokine levels were measured from serum collected before and 1.5 hours after transfusion of LE-WB. Three chemokines were broadly increased after transfusion (Fig 4A). Another 6 cytokines were increased in wild-type mice but to a lesser extent, or not at all, in immunodeficient mice (Fig 4B). Notable among these was the observed increases in the pro-inflammatory-cytokines monocyte chemoattractant protein (MCP)-1, tumor necrosis factor (TNF)α, keratinocyte-derived chemokine (KC or CXCL1), and interleukin (IL)-6. Whereas most cytokines increased or were unresponsive to transfusion, LIX (C-X-C motif 5) chemokine was decreased in immunodeficient mice carrying human cytokine transgenes (Fig 4C). Another four cytokines demonstrated greater responses in some wild-type mice, indicating differential cytokine responses among the three strains of wild-type mice (Fig 4D). Other cytokines measured were unaffected by transfusion (S1A Fig) or were near (S1B Fig) or below the levels of accurate detection (S1C Fig).

## Transfused human plasma does not enhance the clearance of human cells in NSG mice

Most experiments described thus far used donor cells prepared from buffy coats that included a portion of the plasma containing platelets, as well as red cells from below the leukocyte

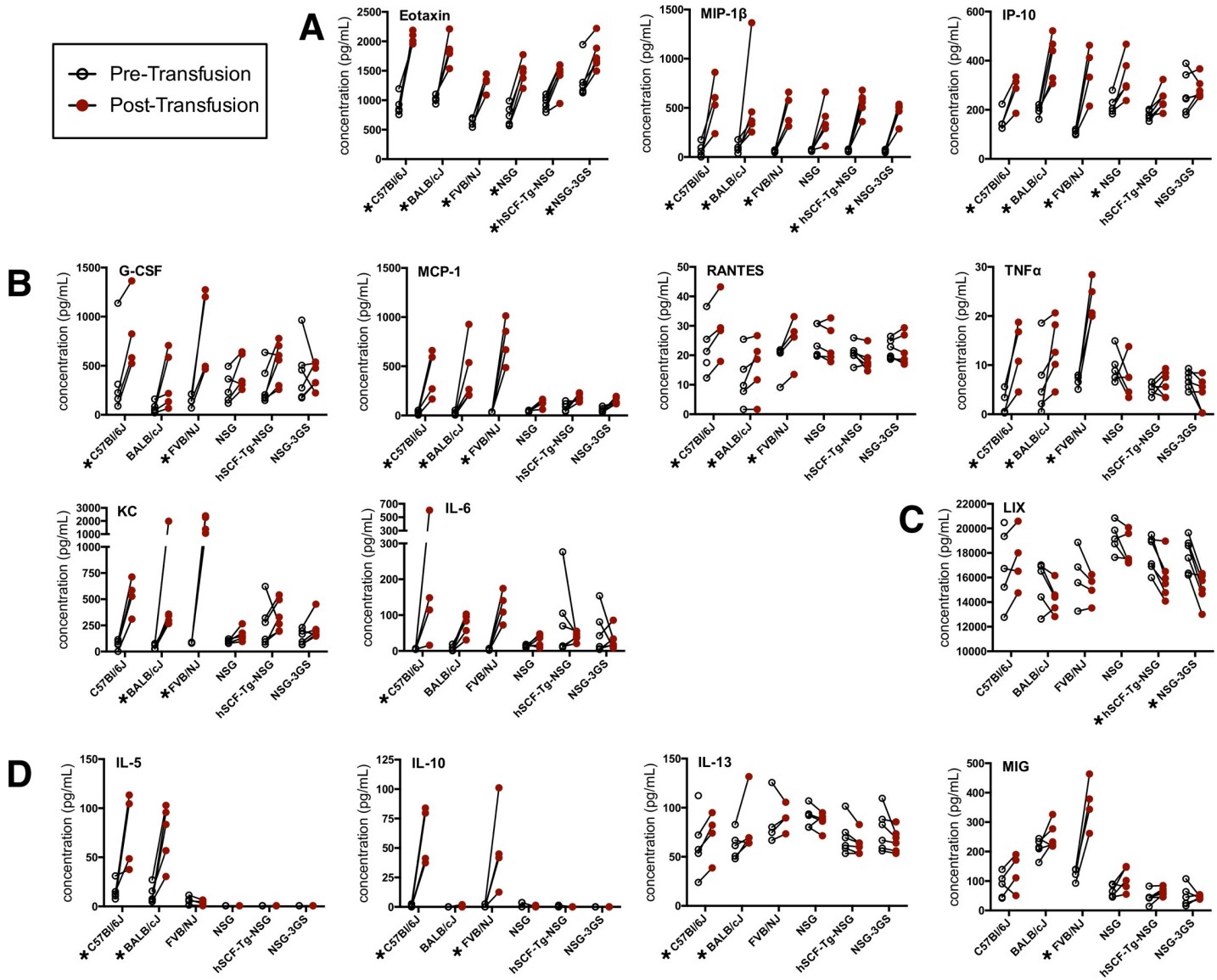

**Fig 4. Cytokine levels in the serum of mice before and 1.5 hours after transfusion with human blood.** (**A**) Cytokines increased by transfusion: Eotaxin, MIP-1β, and IP-10. (**B**) Cytokines increased by transfusion in wild-type mice: G-CSF, MCP-1, RANTES, TNFα, KC, and IL-6. (**C**) LIX was the only cytokine found to be decreased by transfusion in two immunodeficient strains. (**D**) Cytokines with notable differential responses among wild-type mice and between wild-type and immunodeficient mice: IL-5, IL-10, IL-13 and MIG. Significant differences between pre- and post-transfusion are indicated by an asterisk by the name of the mouse strain. The data are from samples taken from 5 B6, 5 BALB/c, 4 FVBN, 5 NSG, 6 hSCF-Tg-NSG (hKL-NSG), and 6 NSG-3GS (NSG-3GMS) mice.

fraction (Table 1). This leukocyte- and platelet-enriched whole blood, LE-WB, provided sufficient numbers of each cell type for evaluation after transfusion, but also resulted in the transfusion of human plasma unless the cells were washed prior to transfusion. To better understand any potential effects of human plasma on the clearance of human cells after transfusion, we performed experiments using washed LE-WB cells suspended in their own plasma or in PBS. Note that total platelet numbers tended to be less in preparations of washed LE-WB cells (Table 1). Overall, higher numbers of human cells were recovered from mice receiving cells with plasma (Fig 5A). The rebound effect observed in earlier experiments was less conspicuous but was still observed with leukocytes and less so with platelets.

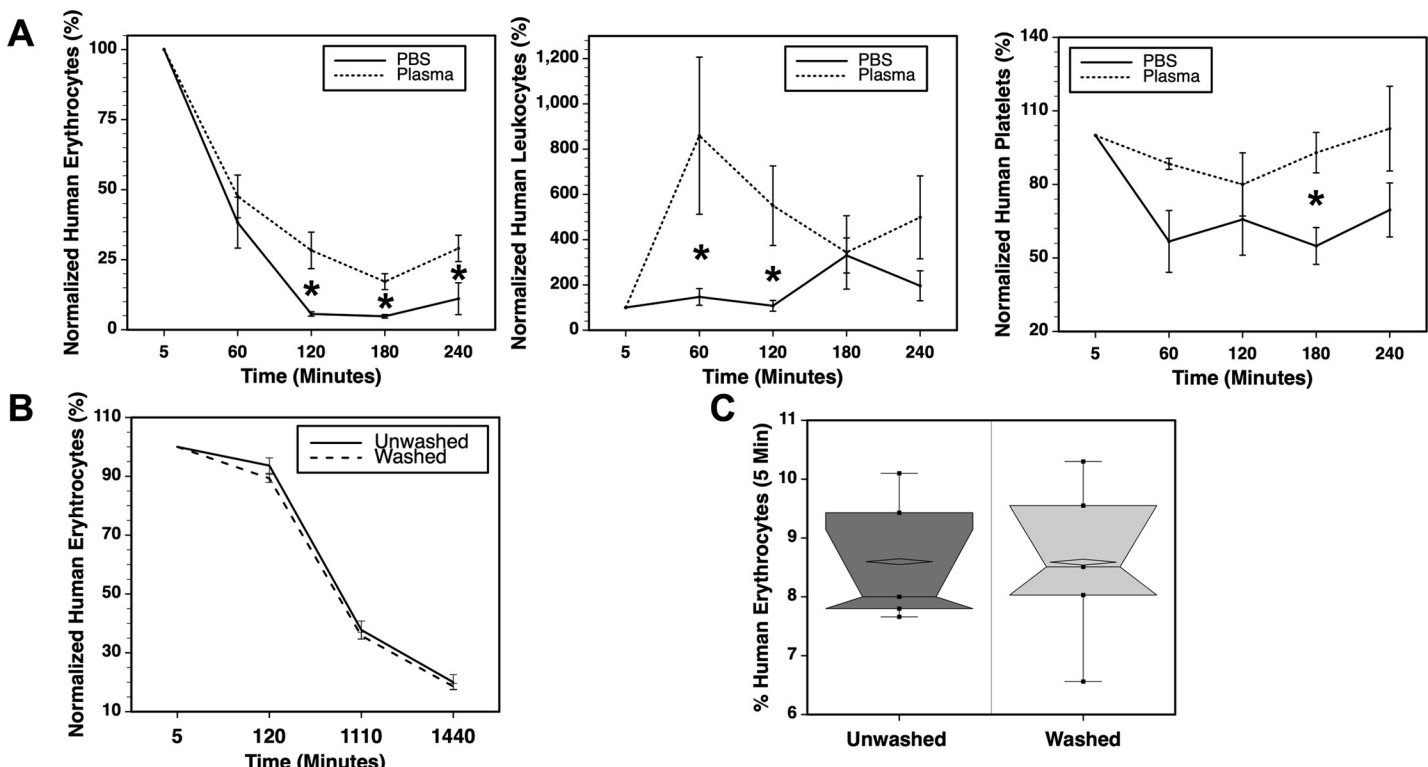

**Fig 5. Effects of human plasma on cell clearance in NSG mice.** (**A**) Mice were transfused with washed LE-WB diluted with either PBS or autologous plasma. (**B**) NSG mice were transfused with washed and unwashed LR-pRBCs. (**C**) Comparison between the initial measured frequency of erythrocytes in mice transfused with LR-pRBCs. In both experiments, 5 mice were transfused per group.

We further tested LR-pRBCs, including a fraction of these cells washed thrice to remove any remaining plasma (Table 1). There were no differences in the frequencies of circulating washed- and unwashed-erythrocytes over a 24-hour period (Fig 5B). The raw frequencies at the initial time point were also similar for both groups (Fig 5C). Thus, there was no indication that washing of LR-pRBCs to remove any residual plasma had any effect on the survival of the cells in NSG hosts.

## Cell dose but not recipient sex affects the clearance of human red cells

The effect of cell dose was examined by transfusing two doses of packed red cells. The frequencies of erythrocytes measured showed greater differences than expected based on the dose administered (Table 1), which is evident when the data are not normalized (Fig 6A). The group receiving the half-lower dose had <25% of the erythrocytes circulating after 10 minutes. This trend continued throughout the experiment. The frequencies of normalized erythrocytes were also significantly lower at 4 and 24 hours in the low dose group pointing towards a more rapid clearance in the low dose group.

The possible role played by the sex of the recipient animals was also analyzed among high and low dose recipients. At the high cell dose there was no significant difference in chimerism between the sexes except for at the last time point at 24 hours (Fig 6B). At the lower cell dose, when clearance mechanisms in the mice are less taxed, recipient sex had no significant effects on normalized donor cell frequencies.

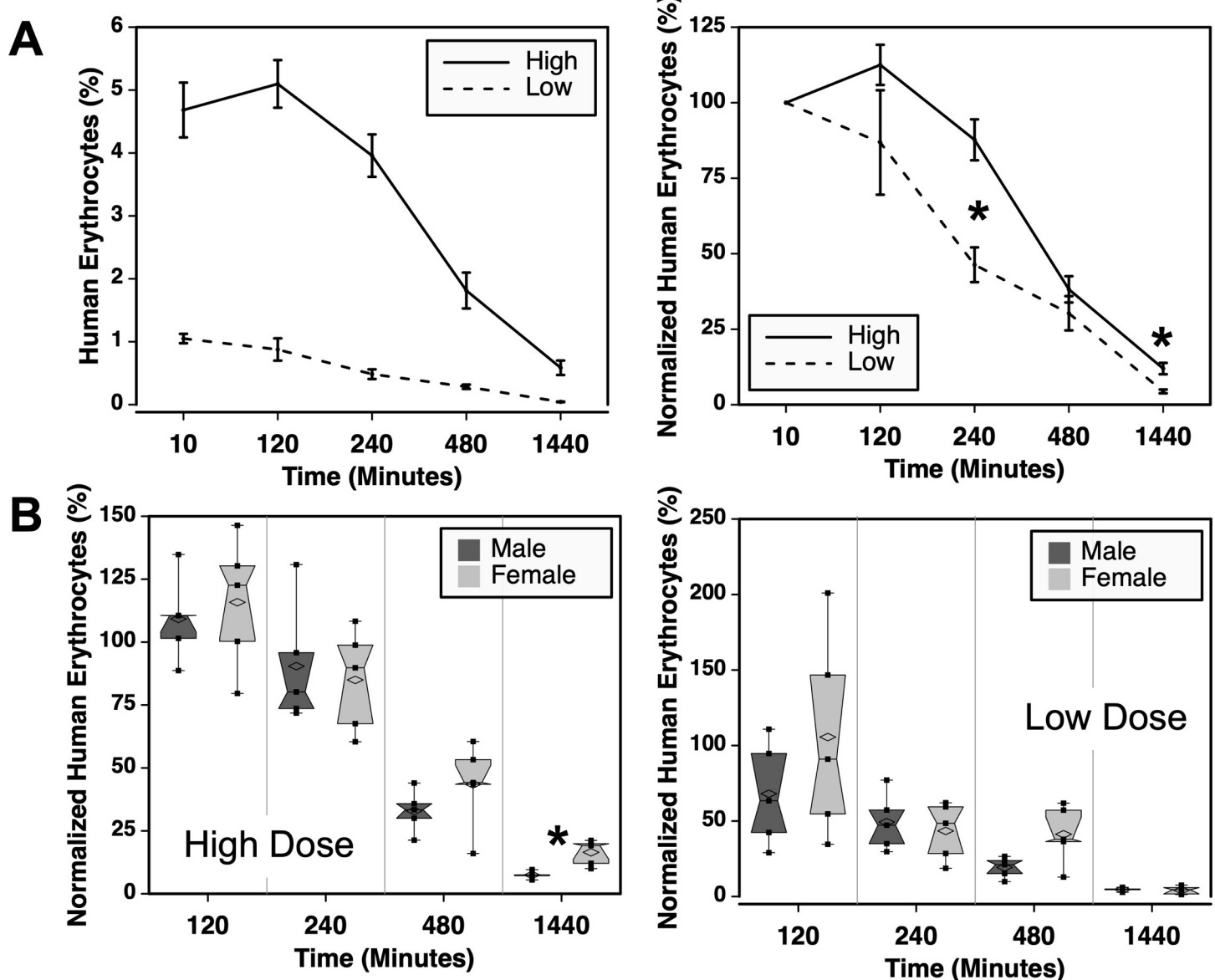

**Fig 6. Effects of cell dose on RBC clearance in NSG mice.** (**A**) Comparison of the frequencies and normalized frequencies of erythrocytes in mice transfused with two doses of LR-pRBCs. Ten mice were transfused with each cell dose, consisting of 5 male and 5 female mice. (**B**) Sex of the recipient did not notably affect normalized erythrocyte frequencies. Asterisks indicated P ≤ 0.05.

## Xenotransfusion leads to a mild and transient leukocytosis in NSG mice

The blood of recipient NSG mice was analyzed over the course of 24 hours after transfusion with LR-pRBCs. Human erythrocytes are larger than mouse erythrocytes and the differences were detectable on the complete blood count (CBC) analyzer shortly after transfusion (Fig 7A) [20]. Despite being able to observe measurable numbers of human cells, the overall effects of transfusion on red cell parameters were negligible with the exception of mean cell hemoglobin (MCH) and mean corpuscular hemoglobin (MCHC) concentration which were elevated above normal ranges after transfusion (Fig 7B). Platelet concentrations made a notable dip within the first hour after transfusion followed by a recovery (Fig 7C). Leukocyte numbers,

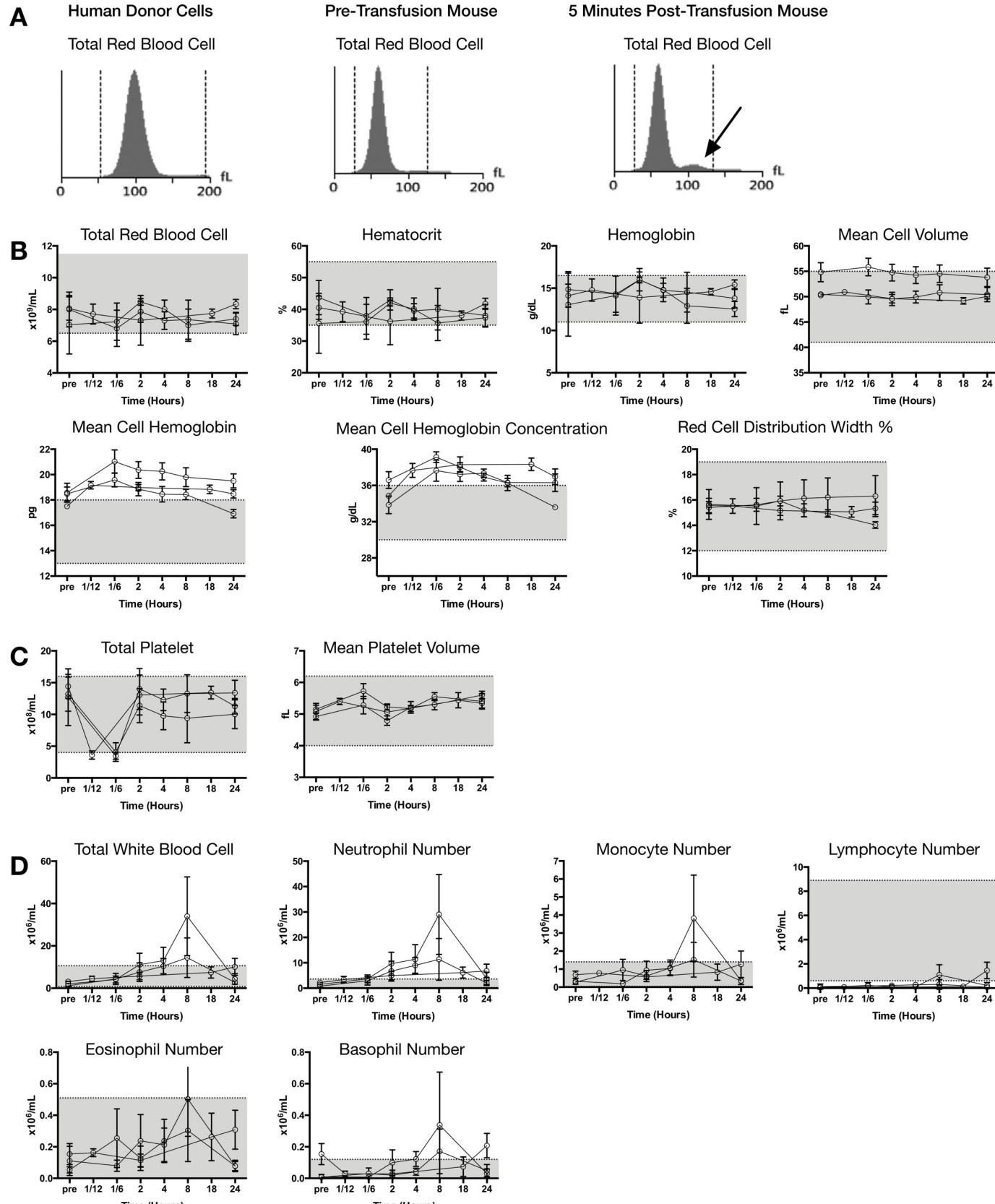

**Fig 7. Blood cell counts in transfused NSG mice.** (**A**) Size comparison of human red blood cells and mouse blood before transfusion and after transfusion indicating (arrow) larger human erythrocytes. Hematological parameters related to red cells (**B**), platelets (**C**), and leukocytes (**D**) are shown for 3 independent experiments (n = 5–10 mice each). Measurements were made before and at various time points after transfusion, differing for the 3 experiments, up to 24 hours after transfusion. Shaded areas represent the normal range for immune-competent mice as reported by the hematology analyzer manufacturer.

comprised mainly of neutrophils and monocytes as NSG mice have no lymphocytes, increased notably around 8 hours after transfusion, but subsided to mostly normal levels 1 day after transfusion (Fig 7D).

## Damaged human RBCs are cleared more rapidly in NSG mice

Reproducibility and sensitivity to differences in blood component quality were examined by transfusing differently treated LR-pRBCs (Fig 8). The donor cells used for this experiment were 21-day old unexpired LR-pRBC (Table 1) obtained from the same unit that was transfused 7 days after donation as shown in Fig 6. This gave the opportunity to evaluate blood from the same donor transfused into mice at two weeks apart. The kinetics of cell clearance were similar for both groups although significantly higher numbers were observed after the initial 10-minute sample and at three time points with the younger, 7-day cells (Fig 8A). However, the two groups of NSG mice that were used did differ in weight. The mean weight of the mice receiving the 7-day cells was 22.9 g (range 20–30 g, n = 10) whereas the recipients of the older cells weighed a mean 33.4 g (range 32–35 g, n = 5). Thus, the larger sizes of the recipients of the 21-day cells likely accounted for the modest, but significant, differences in the frequencies of human erythrocytes.

The 21-day old LR-pRBCs were either untreated or exposed to one or two rounds of γ-irradiation and their clearance in NSG mice compared. In this case, the weights of the recipients were closely matched. Recipient mice of the irradiated sample had the same mean weight (33.4 g, range 31–36 g, n = 5) as the 21-day cell recipients. Mice receiving the twice-irradiated cells had a mean weight of 35.4 g (range 32–40 g, n = 5). Overall, the effects of γ-radiation were minimal on the kinetics of clearance of the LR-pRBCs (Fig 8B). No significant differences were observed compared to the 21-day cells. Significant differences were observed between the two radiation treated groups at the last two time points. Overall, the data in Fig 8B show that consistent results can be obtained by transfusing similar numbers of cells into mice of similar weight.

To further test the ability to detect differences in the quality of erythrocytes in NSG mice, LR-pRBC were treated with diamide to induce oxidative stress [13]. The weights of the recipient mice were similar with the untreated control group mean weight being 25.8 g (range 23–29 g, n = 4) compared to the recipients of diamide treated blood with a mean weight of 26.0 g (range 23–28 g, n = 4). The initial frequency of human erythrocytes measured at 10 minutes after injection was similar for both groups (Fig 8C). Two hours after transfusion the chimerism values remained similar but then significantly diverged by the 4 hour time point. Thereafter, diamide treated RBC cleared more rapidly from the circulation of NSG mice.

## Rapid clearance of human RBC in the first half an hour after transfusion

For the experiments presented, an initial measurement was made at an early time point and used to normalize subsequent measurements for variables such as the differences in blood volumes of the recipient mice. These initial measurements were taken between 5 and 20 minutes, depending on the number of mice transfused and other logistical considerations. To better understand the early dynamics of circulating donor cells, changes in the frequencies of circulating human erythrocyte in the first half an hour after transfusion were analyzed in NSG mice (Fig 9). In one experiment, a decline was observed in the frequency of human cells over the

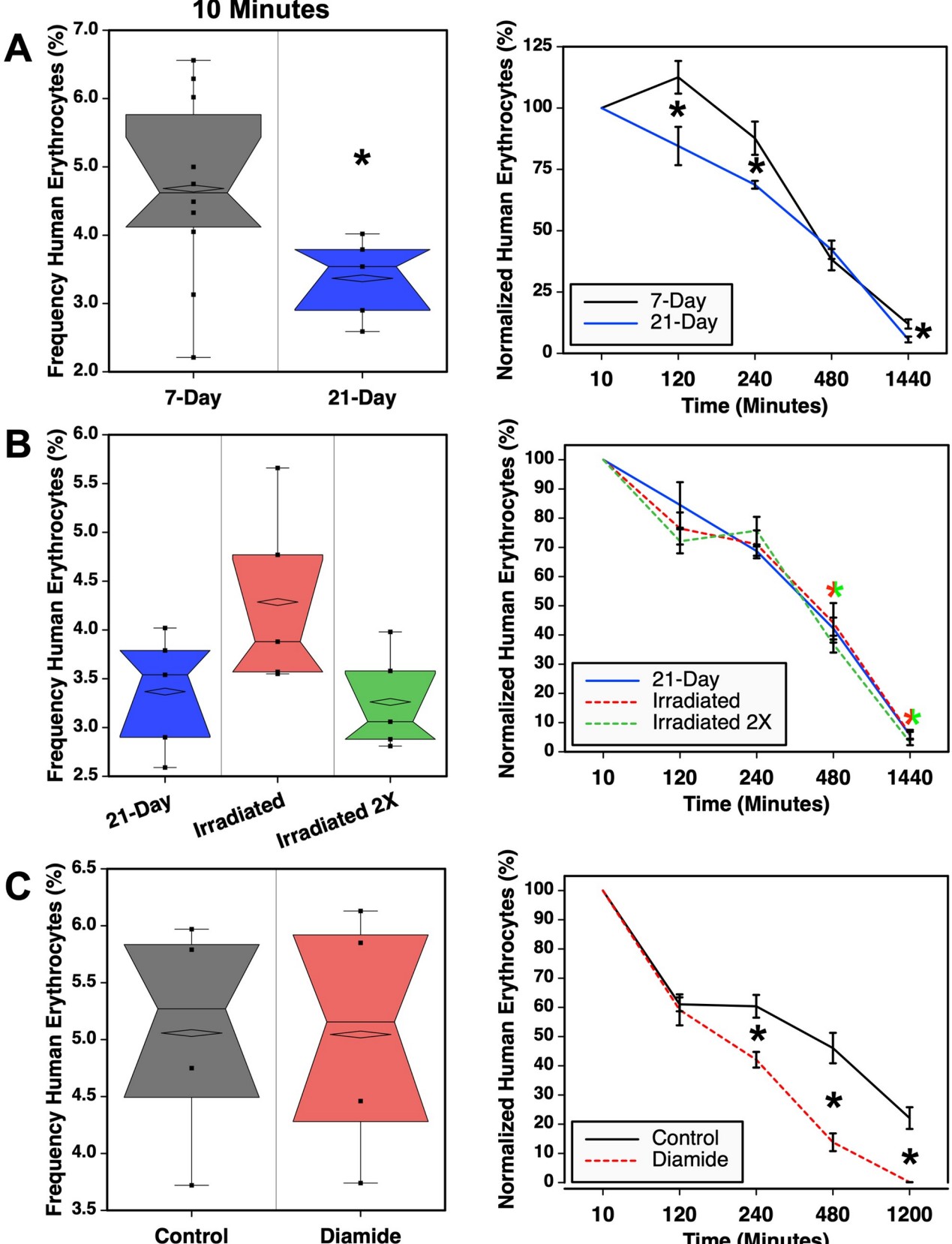

**Fig 8. Effects of age, radiation, and chemical damage on RBC clearance.** (**A**) The same specimen of LR-pRBC was transfused 7-days and 21-days after donation into n = 10 (5 male and 5 female) and n = 5 male NSG mice, respectively. Note, that the data for the 7-day sample are the same as the

high dose sample shown in Fig 6. **(B)** Comparison of erythrocyte clearance for three treatments of LR-pRBCs. The same 21-day sample shown in **(A)** was exposed to one or two rounds of γ-radiation treatment and the blood was transfused into groups of 5 male mice. Colored asterisks indicate P ≤ 0.05 between the two irradiated samples. **(C)** The effects of diamide treatment of RBC on post-transfusion survival are shown in groups of 4 mice. Box plots in the left column show the frequencies of erythrocytes at the initial 10-minute time point. The right column shows normalized data. All asterisks indicated P ≤ 0.05.

first 20 minutes after transfusion (Fig 9A). Although the differences were not significant between 5 and 10 minutes, after 20 or 30 minutes the frequencies of human cells were significantly less than at either 5 or 10 minutes (P < 0.05). Fig 9B shows these early measurements in the context of subsequent measurements made over a 24-hour period. These data were normalized using the 5, 10, and 20 minute measurements (Fig 9D) showing that using the 5 and 10 minute measurements yielded curves most similar to the raw frequency data (Fig 9B).

Another experiment was performed examining human erythrocyte frequencies within the first 5 minutes after transfusion (Fig 9C). The highest mean and median frequencies were observed at 2 minutes after injection, although there were no significant differences among the measurements made between 1 and 4 minutes. Measurements made between 1 and 3 minutes were significantly higher than the 5 minute measurements (P≤0.021). Thus, the distribution of human cells after injection occurs rapidly allowing for measurements of chimerism to be made within the first few minutes to obtain maximum values.

The effects of recipient weight on the initial frequency of human cells was analyzed revealing no clear association between weight and the frequencies of RBCs, leukocytes, and platelets

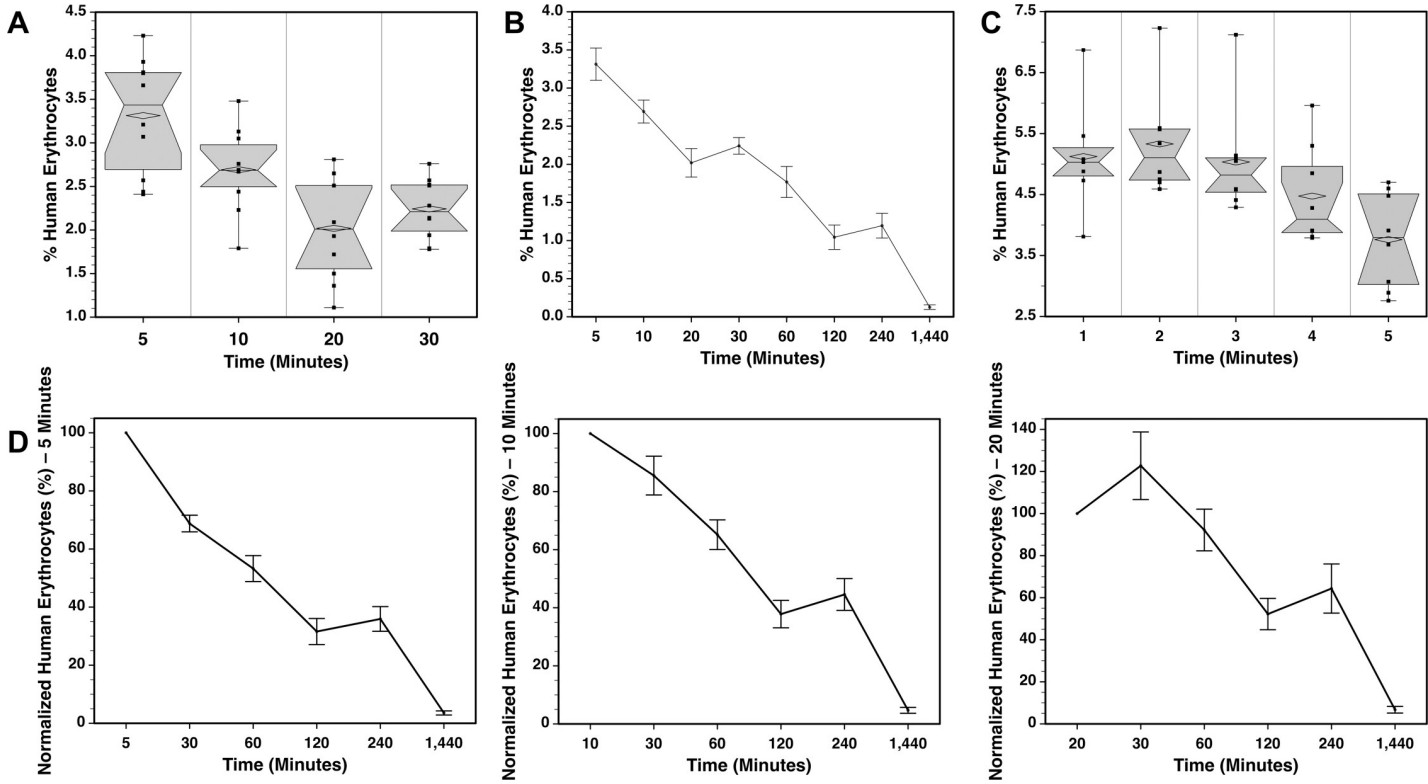

**Fig 9. Rapid clearance of human LR-pRBC. (A)** Box plots of human erythrocyte chimerism measured 5–30 minutes after transfusion (n = 10 male NSG mice). **(B)** Measured frequencies of human erythrocytes 5 minutes to 1 day after transfusion. **(C)** Box plots of human erythrocyte chimerism measured 1–5 minutes after transfusion (n = 8 NSG mice, 4 male and 4 female). **(D)** Frequency data shown in **(A)** and **(B)** plotted in three charts by normalizing using measurements made at 5, 10, or 20 minutes (across, respectively).

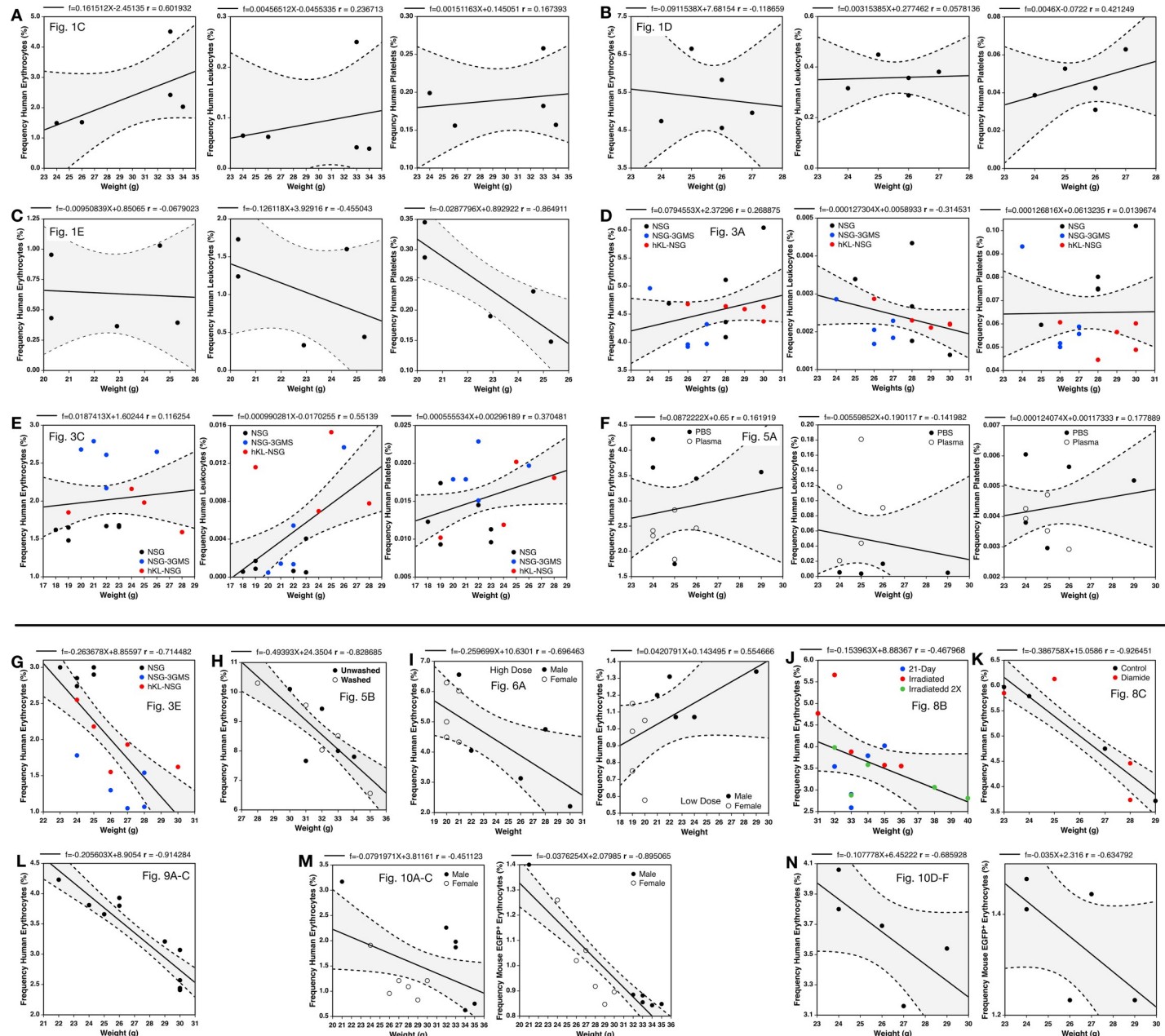

**Fig 10. Relationships between mouse weight and the initial frequencies of human cells.** Each lettered subsection represents data from a single experiment. Subsections (**A-F**) were from experiments in which mice were transfused with LE-WB. These experiments are separated above by a solid line from the below subsections (**G-N**), which were derived from mice transfused with LR-pRBC. All chimerism measurements were made at 5 minutes with the exception of (**I-K**) which were measured at 10 minutes after transfusion. Each experiment is also labeled with the corresponding figure in which the data are first presented. Linear regression was used, with formulas and r-values shown above each graph, and 85% confidence intervals are shown (shaded regions). Note, the frequencies of the co-transfused human and mouse EGFP⁺ erythrocytes are shown in separate graphs in (**M and N**).

in mice transfused with LE-WB (Fig 10A–10F). However, there was, in most cases, a correlation between a smaller body size and a higher initial human cell frequency in mice transfused with LR-pRBC (Fig 10G–10N). An interesting exception to these findings with erythrocyte transfusions was the transfusion of a low dose of cells, but not high dose transfusions, shown in Fig 10I.

Given the relationship between the size of mice and the levels of chimerism achieved, the validity of data normalization based on an initial frequency measurement was evaluated. An analysis of the relationships between the standard deviations and means at each time point for the raw frequency data, shown in Fig 9B, and data normalized using the 5 minute time point (Fig 9D) showed an average reduction in the standard deviation values from 38.0% of the mean to 34.8% of the mean in the normalized data. Thus, frequency normalization may help to modestly reduce variability that arises from differences in the size of recipient blood volume as well as other experimental variables such as the precision of the injections.

To further understand the fate of transfused human erythrocytes in the immediate minutes following injection, we transfused human RBCs into NSG mice in combination with mouse RBCs obtained from BALB/c mice expressing the EGFP fluorescent protein. Note, the NSG-EGFP line of mice was deemed not suitable for this experiment as only nucleated cells expressed EGFP. In comparison to the human cells, the transfused mouse cells were not rapidly cleared from the circulation in two experiments (Fig 11). Indeed, at the 5 minute measurement, the frequency of human cells was higher than the transfused mouse cells, but after 1 hour the frequency of human cells had fallen below the mouse cells (Fig 11B). Given the subsequent stability in the frequencies of the EGFP$^+$ RBCs, these data indicate removal of human cells from the circulation within the first 5 minutes after transfusion.

The ratio of human to mouse cells was determined for the mixture of donor cells, as well as for all the measurements made after transfusion, indicating a rapid decline in the value before the first measurement at 5 minutes (Fig 11C). In the second experiment that examined time points before 5 minutes, the ratio of human to mouse cells 1 minute after transfusion did not significantly differ from the subsequent measurements until the 5 minute (P = 0.0026) time point (Fig 11F). Two minutes after transfusion, the ratio was similar to those at the 1 minute, and differed significantly to those at 4 and 5 minutes after transfusion (P ≤ 0.0274). These data further highlight a brief period of stability during the first few minutes after transfusion marking the peak numbers of human cells in the circulation.

## Clearance of human RBC from the circulation within 2 days

The longevity of human cells in the circulation was also analyzed in the two experiments shown in Fig 11. In the first experiment the average frequency of human erythrocytes was 0.0418 ±0.0192% one day after transfusion and 0.0005±0.0001% after 2 days (Fig 11B). In the second experiment, which started with a higher frequency of human RBC 5 minutes after transfusion, the mean frequency was 0.0930±0.0416% after 1 day (Fig 11E). At 2 and 3 days, the levels of human chimerism fell below the limits of detection (0.0043%) based on an analysis of an untransfused mouse. These data are consistent with overall findings that low frequencies of human cells may be detected for at least 1 day after transfusion e.g., Figs 1E, 3, 5B, 6, 8 and 9.

The half-life of human erythrocytes was estimated based on data from NSG mice transfused with LR-pRBC (S2 Fig). Using linear regression, the mean half-life was 490 minutes for 9 samples, including samples with different doses of cells, treated with radiation, or diamide (Table 2). A logarithmic regression was also examined, which gave a lower 249 minute half-life, and better fit the data in some experiments (S2 Fig) as well as more closely matched direct measurements made nearest the calculated half-life (Table 2).

## Human erythrocytes accumulate in the lungs and liver shortly after transfusion

The tissue distribution of human red blood cells was evaluated 5–6 minutes after transfusion (Fig 12). Five minutes after transfusion a small blood sample was taken for flow cytometric

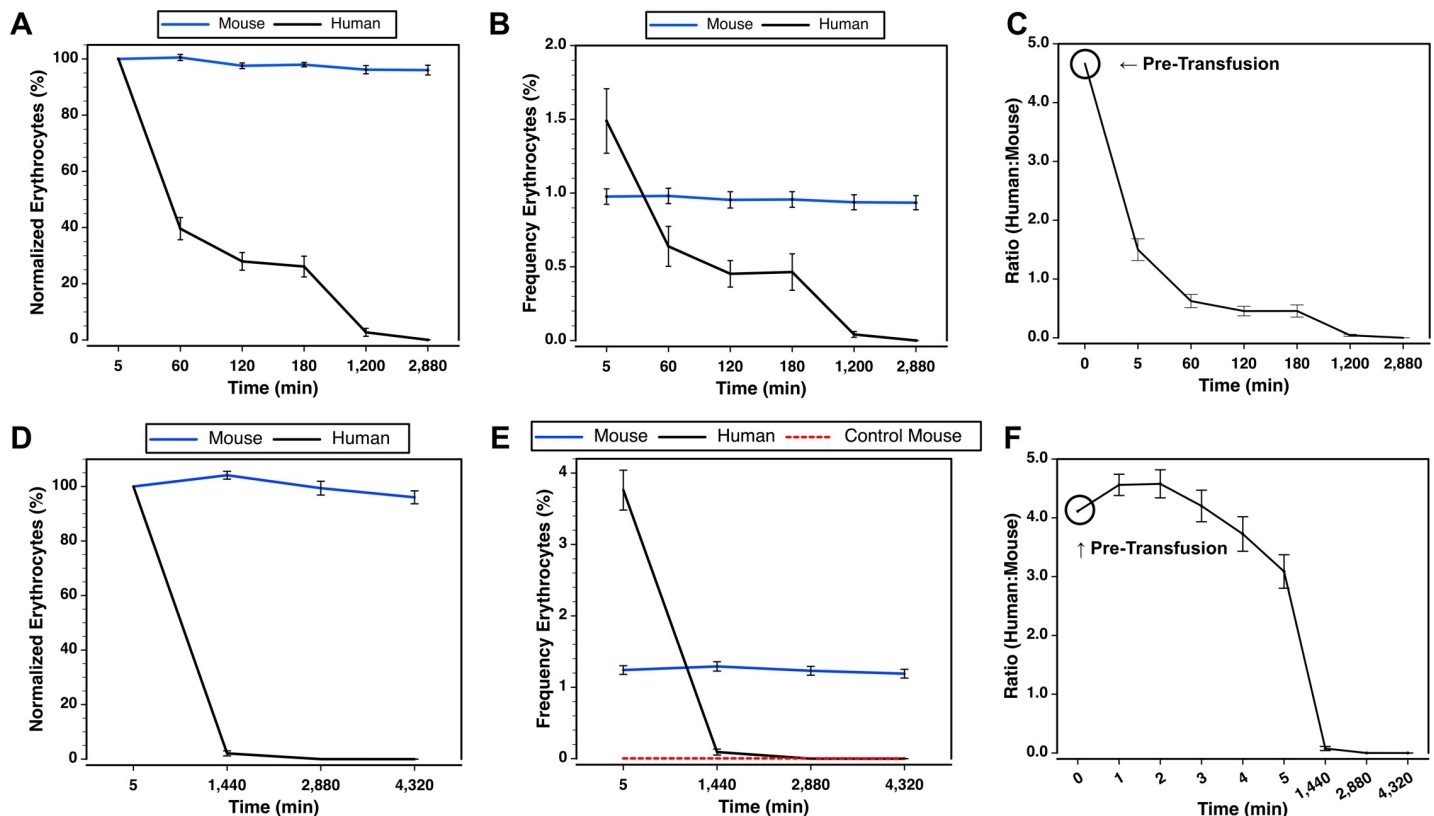

**Fig 11. Comparison of cell clearance of co-transfused mouse and human erythrocytes.** Two experiments were performed in NSG mice by transfusing n = 6 female and 6 male mice (**A-C**), and n = 4 male and 4 female mice (**D-F**). Note, data from the second experiment are also shown in Fig 9C. (**A and D**) Comparison of normalized erythrocyte frequencies of human and EGFP⁺ mouse erythrocytes co-transfused into NSG mice and analyzed between 5 minutes up to 3 days after transfusion. (**B and E**) Comparison of the measured frequencies of co-transfused human and mouse erythrocytes. (**E**) Background levels are shown using a red line based on analysis of an untransfused control NSG mouse. (**C and F**) The ratios of human to mouse EGFP⁺ erythrocytes before and after transfusion.

analysis followed by a larger sample for CBC analysis in a process that lasted about 30 seconds to a minute. Thereafter, the mice were immediately sacrificed and tissues harvested and processed for analysis by flow cytometry (Fig 12A). The co-transfused mouse blood cells were used as a reference for normal tissue distribution with the highest frequencies observed in the blood (Fig 12B), followed by the liver, lung, and then spleen. There was no difference between

**Table 2. Half-life of human erythrocytes in the circulation of NSG mice.**

| Experiment | Logarithmic | Linear | Nearest Time Point to Half Life (Mean Observed % Normalized RBC) |
|---|---|---|---|
| Fig 6A—High Dose | 553 minutes | 756 minutes | 480 minutes (36.5%) |
| Fig 6A—Low Dose | 218 minutes | 518 minutes | 240 minutes (46.4%) |
| Fig 8A—21 Day | 318 minutes | 624 minutes | 480 minutes (42.2%) |
| Fig 8B—Irradiated | 311 minutes | 618 minutes | 480 minutes (44.2%) |
| Fig 8B - 2X Irradiated | 292 minutes | 588 minutes | 480 minutes (36.9%) |
| Fig 8C—Control | 294 minutes | 561 minutes | 480 minutes (46.1%) |
| Fig 8C—Diamide | 127 minutes | 311 minutes | 240 minutes (42.1%) |
| Fig 9D—5 minutes | 73.7 minutes | 346 minutes | 60 minutes (53.3%) |
| Fig 11A | 52.3 minutes | 86.2 minutes | 60 minutes (39.6%) |
| **Mean** | **249 minutes** | **490 minutes** | |

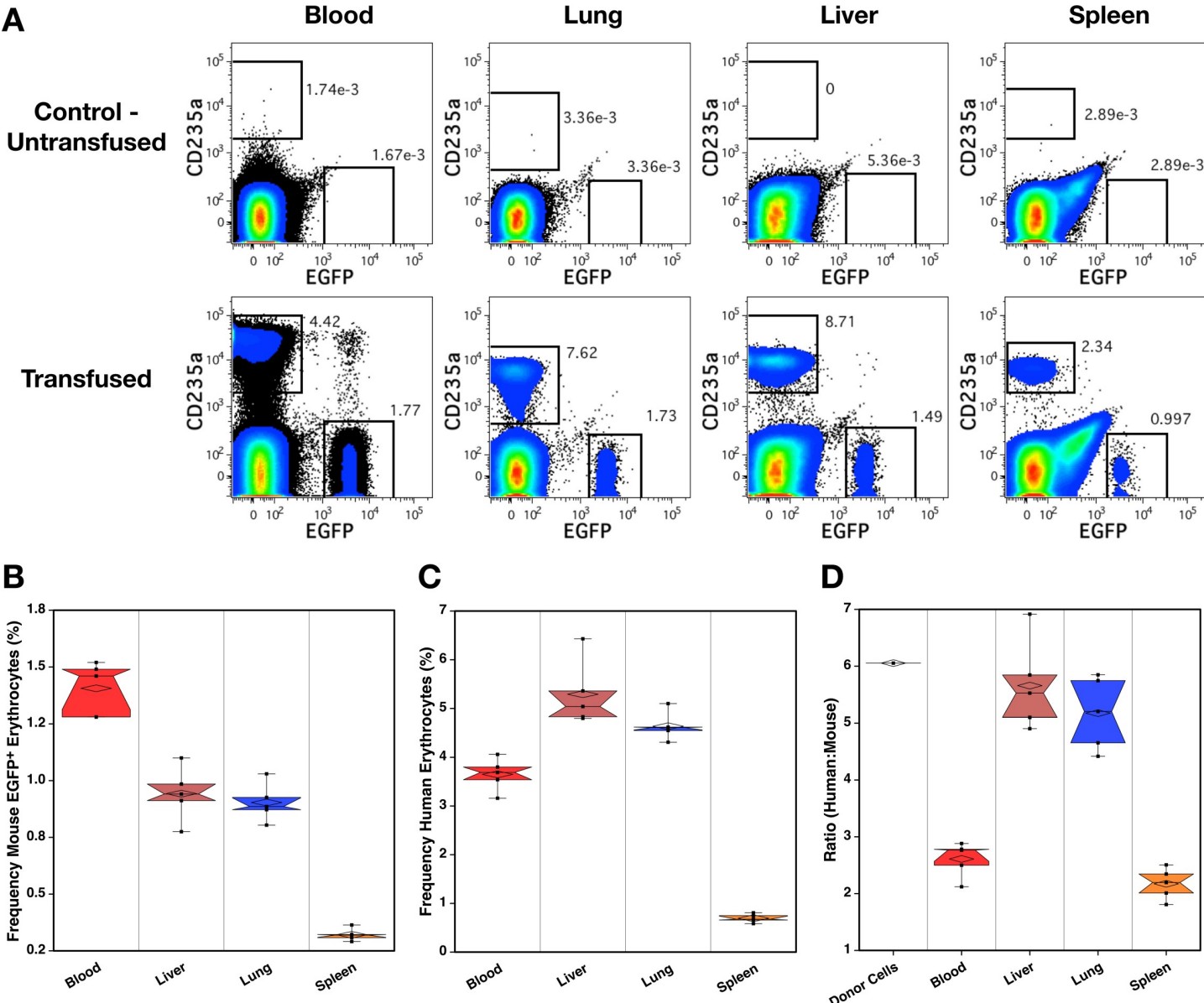

**Fig 12. Tissue distribution of human red blood cells shortly after transfusion. (A)** Representative flow cytometric analyses of different cell compartments 5 minutes after transfusion as well as a control untransfused mouse. Frequencies of transfused EGFP+ mouse cells **(B)** and human red blood cells **(C)** in the blood liver, lung and spleens. **(D)** The ratio of human to mouse EGFP+ erythrocytes before (Donor Cells) and after transfusion in the blood and indicated tissues.

the frequencies in the liver and lung, whereas all other differences were significant (P ≤ 0.05). The distribution of human cells (Fig 12C) differed from those of the transfused mouse cells in that higher frequencies were observed in the liver and lungs than in the blood (P = 0.009). Moreover, the frequencies of human cells in the liver were higher than in the lungs (P ≤ 0.0472). Just as for the transfused mouse cells, the spleen contained the lowest frequency of human cells. When these data are viewed as ratios of human to mouse cells (Fig 12D), the similarities in the ratios of these cells in the blood and spleen are apparent as well as the selective accumulation of human cells in the liver and lung (P = 0.009).

Two approaches were taken to quantify the distribution of human cells. Based on the dilution of a known quantity of transfused EGFP$^+$ mouse erythrocytes, the mean estimated blood volume was 1008 μl for NSG mice (n = 5) weighing a mean 26.0 g (38.8 μl / g). A mean 43.1% of all human erythrocytes were estimated to be in circulation using this method. The mean content of human cells in the liver, lung, and spleen was 0.12%, 0.034%, and 0.001% of the total transfused RBCs, respectively. Fully 56.7% of transfused human RBCs were not accounted for in these measurements. In a second approach, blood volume was estimated as by Sluiter et al. [17], resulting in a higher mean volume of 1791 μl for the 5 transfused NSG mice. The higher blood volume estimates leads to a higher mean percentage of human cells in the circulation of 76.6%. Using either of the methods for estimation of blood volume, far more human erythrocytes were found in the blood than in the organs analyzed.

## Discussion

Xenogeneic transfusion models in mice are essential to studying various aspects of blood component quality or transfusion reactions. A concern with the value of such models is the relatively short lifespan of human blood components in the mouse circulation. This study was aimed at better understanding and refining a mouse model by evaluating the recoveries of human blood cells transfused into multiple mouse strains of different genetic backgrounds. The rapid clearance of human cells in wild-type mouse strains suggests that innate immune mechanisms are largely responsible for the rapid clearance of human cells. Therefore, immunodeficient NSG mice, having adaptive and innate immune deficiencies, proved to be advantageous in modeling human transfusion. Immunodeficient NSG mice offer a clear survival advantage for human erythrocytes, platelets and leukocytes compared to the wild-type strains of mice that were tested. Nonetheless, clearance of human cells from the circulation of NSG mice is still a rapid event affected by the dose and types of cells administered. The mechanisms of human cell clearance from the circulation appear multifaceted with the hemolytic complement defect in NSG mice playing a role in the prolonged survival of human cells [4].

The presence of hemoglobin in the urine coupled with notable cytokine responses are indicative of the occurrence of an acute hemolytic transfusion reaction, which was pronounced in wild-type mice. Xenotransfusion in wild-type mice, but not immunodeficient mice, resulted in elevated levels of MCP-1, KC (CXCL1), IL-6, and TNF-α that are consistent with reports on hemolytic transfusion reactions in mice and human patients [14, 21, 22]. These elevated cytokine levels, in particular the inflammatory cytokine and pyrogen TNF-α, were also the likely cause of the lethargy observed in wild-type mice in the immediate hours following transfusion. We also observed increased levels of G-CSF, RANTES, IL-5, IL-10, IL-13, and MIG in at least some wild-type mouse strains compared to immunodeficient mice. Many of these changes are consistent with previous observations of early responses to traumatic blood loss and allogeneic transfusion; elevated levels of MCP-1, KC, IL-5, IL-6, IL-10, and IL-13 were observed in response to blood loss, whereas only MCP-1 and IL-5 increased in response to allogeneic whole blood transfusion alone [16]. Thus, the cytokine response observed in xenotransfused wild-type mice more closely resembles that of a traumatic injury than a response to allogeneic transfusion.

The transfusion of leukocyte-enriched whole blood into immunodeficient mice resulted, to varying degrees, in a rebound effect characterized by higher frequencies of human blood cells a few hours after transfusion than measured at the reference time point shortly after transfusion. This effect was diminished when donor cells were washed prior to transfusion or when LR-pRBCs were transfused. Shortly after transfusion, most LR-pRBCs were found in the circulation, although a large portion–varying with the method used to estimate blood volume–of

the transfused cells were unaccounted for. Measurement errors could account for some of the missing human cells. No rebound effect was observed in previous xenogeneic transfusion studies using leukoreduced apheresis platelets [23, 24]. Thus, the transfusion of leukocytes, or perhaps the combination of different human blood cells, appears to contribute to events affecting the circulation of human cells. The rebound effect points to a sequestration of the human cells followed by their release. Human blood cells may temporarily adhere to the vasculature or be sequestered in tissues as a result of xenotransfusion, as was recently shown to occur with autologous transfusion in humans [25]. Disseminated intravascular coagulation, as a result of interspecies incompatibilities in coagulation factors, may also play a role in the rebound effect [26]. Indeed, hemolytic transfusion reactions, as observed to some degree even in immunodeficient mice, can result in disseminated intravascular coagulation [27]. Heme can cause platelet activation and aggregation [28], and we observed a decrease in overall platelet levels 30 minutes post-transfusion which increased ~2 hours post-transfusion, consistent with platelet adhesion and aggregation. Most importantly for the use of NSG mice to model the transfusion of human cells, it appears that the rebound effect is a short-term phenomenon that can be avoided if blood components are used rather than whole blood.

One of the differences between the bm12, B6, and BALBc mouse strains tested and the FVBN and NSG strains is a deletion of the C5 gene [29, 30]. The findings of hemoglobinuria and the observed cytokine responses in wild-type mice are consistent with an intravascular hemolytic transfusion reaction being responsible for the rapid clearance of human cells [31]. Rapid cell destruction is also the likely explanation for the lack of a rebound effect in wild-type mice as the human cells are eliminated by hemolytic complement activity rather than being sequestered. Of note, FVBN mice provided only a modest survival advantage and cytokine responses were more similar to other wild-type strains than NSG mice in most measures–with IL-5, IL-13, and MIG production being exceptions. A low level of hemoglobinuria was also observed in immunodeficient mice. It is possible that the transfusion of human plasma in FVBN and NSG mice resulted in a partial reconstitution of the complement pathway as human C5 can substitute murine C5 activity [32], but we did not see evidence for increased clearance of human cells in NSG mice when human plasma was transfused with washed human cells. However, immunodeficient mice also lack immunoglobulin, a necessary component of a classical hemolytic transfusion reaction [33].

Immunodeficient mice such as NSG are useful models for human transfusion primarily due to an absence of C5 complement and immunoglobulin, thereby diminishing hemolytic transfusion reactions and associated sequelae. Although human cells are still cleared rapidly in NSG mice, the delays in clearance compared with wild-type strains appear sufficient to allow detection of differences in cell quality as seen with diamide damaged red blood cells. Additionally, we have previously demonstrated measurable differences in survival of transfused platelets that differed by length of storage or storage temperature [23, 24]. These studies also demonstrated a functional effect of the transfused human platelets on vascular endothelial cell permeability.

Another limitation is that through elimination of many of the parts of the immune system responsible for clearance of xenogeneic cells, we may also lose the ability to detect differences in the quality of the transfused cells that are dependent on these clearance mechanisms. For example, depletion of macrophages in immunodeficient mice, using liposomes containing clodronate, increases the survival of human red cells and platelets in the circulation [8, 34]. This shows that macrophages are important mediators of xenogeneic cell clearance. However, macrophages are also important in the physiological removal of old and injured blood cells [35, 36]. Thus, a macrophage-depleted mouse model will likely not accurately measure the quality of human blood products as the cells most critical in the removal of aged or damaged cells

have been severely depleted. However, macrophage depletion of mice is still a useful tool to increase human blood cell chimerism in humanized mice for other purposes.

A number of experimental variables can affect the outcome of human transfusions in NSG mice and are worth considering when using this model. The weight of the mice can affect the initial levels of chimerism, which can be minimized by selecting mice of similar weight or normalization of the data using either an early (2–10 minute) time-point measurements or tracer cells. The sex of the mice did not appear to play a noticeable role in the clearance of human cells, although sex and age can affect the weight of mice. Clearance of human cells from the circulation begins immediately after transfusion with accumulation of human cells in the liver and lungs, which can be measured by co-transfusion of EGFP$^+$ BALB/c cells. The cell dose should also be maintained in a similar range across experiments as lower cell doses lead to more rapid clearance. This observation points to clearance or sequestration mechanisms that can be saturated, albeit briefly. This also has implications for hematopoietic stem cell transplant studies in NSG mice in which different cell doses may have different outcomes not solely due to the content of transplanted stem cells. Mice humanized by stem cell transplants or through transfusion of peripheral blood cells, typically in combination with prior irradiation of the mice, are important model systems. In the absence of irradiation, the use of mice expressing human cytokine transgenes had no clear effect on the immediate survival of leukocytes, unlike what is observed over longer time periods in stem cell transplant models [6, 37]. In conclusion, this work formally demonstrates NSG mice are a superior mouse strain for xenogeneic transfusion studies as they better sustain human cell chimerism without acute transfusion-related side effects observed in wild-type mice.

## Supporting information

**S1 File.**
(PDF)

**S1 Fig. Cytokine levels in the serum of mice before and 1.5 hours after transfusion with human blood.** The indicated strains of mice were transfused with whole blood containing 1.86 x $10^6$ leukocytes, 6.10 x $10^8$ erythrocytes, and 5.54 x $10^7$ platelets. (**A**) Cytokines detected in the sera that were unaffected by transfusion are shown. Cytokines for which only some of the highest measurements were in range of the standard curve are shown in (**B**) and cytokines whose concentration fell bellow accurate detection levels are shown in (**C**). Significant differences between pre- and post-transfusion are indicated by an asterisk by the name of the mouse strain.
(PDF)

**S2 Fig. Half-life determinations for transfused human LR-pRBC.** Each transfused group of NSG mice was analyzed using logarithmic (left) and linear (right) regression. Charts are organized by the corresponding Figure in the manuscript. Any additional sub-groups that define the transfused cell population are indicated in the corners of the charts. The results of half-life calculations using the indicated formulas at the top of each chart are shown in Table 2.
(PDF)

## Acknowledgments

The authors wish to thank Inderdeep Singh and Philip Norris, M.D. for phlebotomy. We also like to thank Kenton Chung, Alvin Hui, Justin Mai, and Karla I. Medina for help with breeding and maintaining the mouse colonies as well as assistance in performing experiments. We also appreciate Orsolya Darst for editing the manuscript. We are also very grateful for the efforts of

James T. Ikeda and the San Mateo High School Biotechnology Internship Program for providing guidance for S.A.B. and J.R.

## Author Contributions

**Conceptualization:** Rachael P. Jackman, Renata Gilfanova, Tamir Kanias, Marcus O. Muench.

**Data curation:** Sophia A. Blessinger, Johnson Q. Tran, Rachael P. Jackman, Jacqueline Rittenhouse, Alan G. Gutierrez, Tamir Kanias, Marcus O. Muench.

**Formal analysis:** Sophia A. Blessinger, Johnson Q. Tran, Rachael P. Jackman, Renata Gilfanova, Jacqueline Rittenhouse, Alan G. Gutierrez, Tamir Kanias, Marcus O. Muench.

**Funding acquisition:** Rachael P. Jackman, Tamir Kanias, Marcus O. Muench.

**Investigation:** Sophia A. Blessinger, Johnson Q. Tran, Rachael P. Jackman, Renata Gilfanova, Jacqueline Rittenhouse, Alan G. Gutierrez, John W. Heitman, Marcus O. Muench.

**Methodology:** Johnson Q. Tran, Rachael P. Jackman, Renata Gilfanova, Tamir Kanias, Marcus O. Muench.

**Project administration:** Rachael P. Jackman, Tamir Kanias, Marcus O. Muench.

**Resources:** Kelsey Hazegh, Tamir Kanias, Marcus O. Muench.

**Supervision:** Rachael P. Jackman, Tamir Kanias, Marcus O. Muench.

**Validation:** Sophia A. Blessinger, Johnson Q. Tran, Rachael P. Jackman, Renata Gilfanova, Alan G. Gutierrez, Tamir Kanias, Marcus O. Muench.

**Visualization:** Sophia A. Blessinger, Johnson Q. Tran, Rachael P. Jackman, Alan G. Gutierrez, Marcus O. Muench.

**Writing – original draft:** Sophia A. Blessinger, Johnson Q. Tran, Rachael P. Jackman, Alan G. Gutierrez, Tamir Kanias, Marcus O. Muench.

**Writing – review & editing:** Sophia A. Blessinger, Johnson Q. Tran, Rachael P. Jackman, Renata Gilfanova, Jacqueline Rittenhouse, Alan G. Gutierrez, John W. Heitman, Kelsey Hazegh, Tamir Kanias, Marcus O. Muench.

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
