## [Decision Letter · Decision Letter 0]

20 Mar 2020

PONE-D-20-05120

Immunodeficient mice are better for modeling the transfusion of human blood components than wild-type mice

PLOS ONE

Dear Dr. Muench,

Thank you for submitting your manuscript to PLOS ONE. After careful consideration, we feel that it has merit but does not fully meet PLOS ONE’s publication criteria as it currently stands. Therefore, we invite you to submit a revised version of the manuscript that addresses the points raised during the review process. In particular, please provide answers to criticism from reviewer #2.

We would appreciate receiving your revised manuscript by May 04 2020 11:59PM. To enhance the reproducibility of your results, we recommend that if applicable you deposit your laboratory protocols in protocols.io, where a protocol can be assigned its own identifier (DOI) such that it can be cited independently in the future. For instructions see: http://journals.plos.org/plosone/s/submission-guidelines#loc-laboratory-protocols

We look forward to receiving your revised manuscript.

Kind regards,

Francesco Bertolini, MD, PhD

Academic Editor

PLOS ONE

Journal Requirements:

2.We note that you have included the phrase “data not shown” in your manuscript. Unfortunately, this does not meet our data sharing requirements. PLOS does not permit references to inaccessible data. We require that authors provide all relevant data within the paper, Supporting Information files, or in an acceptable, public repository. Please add a citation to support this phrase or upload the data that corresponds with these findings to a stable repository (such as Figshare or Dryad) and provide and URLs, DOIs, or accession numbers that may be used to access these data. Or, if the data are not a core part of the research being presented in your study, we ask that you remove the phrase that refers to these data.

Reviewers' comments:

Reviewer's Responses to Questions

**Comments to the Author**

1. Is the manuscript technically sound, and do the data support the conclusions?

Reviewer #1: Yes

Reviewer #2: Yes

2. Has the statistical analysis been performed appropriately and rigorously? 

Reviewer #1: Yes

Reviewer #2: Yes

3. Have the authors made all data underlying the findings in their manuscript fully available?

Reviewer #1: Yes

Reviewer #2: Yes

4. Is the manuscript presented in an intelligible fashion and written in standard English?

Reviewer #1: Yes

Reviewer #2: Yes

5. Review Comments to the Author

Reviewer #1: In this paper, Blessinger et al. compared several strains of immunocompetent wild-type mice of different genetic backgrounds and NSG mice as hosts for transfused human blood cells. As expected, circulating xenogeneic cells were found to be mantained better in NSG rather than in wild-type mice thanks to an immunocompromised enviroment. The authors also investigated the expression of pro-inflammatory cytochines after the xenotransfusion as well as other relevant aspects related to humanization of the hematopoietic system of mice, i.e. any potential effects of human plasma on the clearance of human cells and the transfusion of different donor blood product.

The manuscript is clear, well written and the conclusions are supported by the data. Howerver, in order to make it suitable for publication I strongly encourage the authors to make some changes, more specifically:

In the headings of the result section I suggest to avoid the use of general terms such as "effect" and "roll" and already focus the attention on the main foundings that will be discussed in the paragraph.

The figures must be ameliorated:

- most of them are kinda blurry and apparently submitted at low resolution (for example Fig. 1 and Fig. 3) please provide better quality images.

- verify their readibility; for example, Fig. 4 and Fig. 7 are organized in a confuse way and it is not easy to understand when the various panels start and finish.

- Fig. 2 must be shifted to supporting informations.

Reviewer #2: The paper by "blessinger et al, entitled: immunodeficient mice are better for modelling the transfusion of human blood compoments than wild type mice” aims to study the efficiency of blood cell transfusion in immunodeficient mice. The paper discusses different aspects impacting transfusion efficiency: the dose of blood cells to be transfused, the influence of age and weight of the mice ...

the experiments are well conducted, but questions remain unanswered and papers have already addressed these points in the literature, so this study lacks of novelty.

The questions that should be addressed are the following:

what happens in the first 5 minutes after transfusion? and that is the half-life of each blood cell after transfusion, a follow-up over several days is required.

are the transfused elements functional in these mice?

in addition to transfusion yield expressed as a percentage, what is the approximate number of cells recirculated, this is an important point to address to support the importance of using immunodeficient mice, analyses with count beads used in flow cytometry may address this question.

How is the rebound effect of red blood cells and platelets explained? What is happening? What mechanisms are involved?

the use of clodronate has been discussed in several papers and has been shown to allow better recirculation of the figurative elements of blood without impairing the functionality of the cells, the point should be addressed in the experiments.

6. PLOS authors have the option to publish the peer review history of their article (what does this mean?). If published, this will include your full peer review and any attached files.

Reviewer #1: No

Reviewer #2: No

---

## [Author Response · Author response to Decision Letter 0]

11 Jun 2020

Please note that line numbers refer to the marked-up version of the manuscript, which differs only in respect to line numbering to the clean manuscript due to what appears to be a bug in the MS Word software when applying line numbers to marked-up documents.

Response to editorial office

We note that you have included the phrase “data not shown” in your manuscript. Unfortunately, this does not meet our data sharing requirements. PLOS does not permit references to inaccessible data. We require that authors provide all relevant data within the paper, Supporting Information files, or in an acceptable, public repository. Please add a citation to support this phrase or upload the data that corresponds with these findings to a stable repository (such as Figshare or Dryad) and provide and URLs, DOIs, or accession numbers that may be used to access these data. Or, if the data are not a core part of the research being presented in your study, we ask that you remove the phrase that refers to these data.

We provided all the available data in a new Figure 10 (lines 499-567). These data are discussed in a response to Reviewer 2.

Responses to Reviewer 1

In this paper, Blessinger et al. compared several strains of immunocompetent wild-type mice of different genetic backgrounds and NSG mice as hosts for transfused human blood cells. As expected, circulating xenogeneic cells were found to be mantained better in NSG rather than in wild-type mice thanks to an immunocompromised enviroment. The authors also investigated the expression of pro-inflammatory cytochines after the xenotransfusion as well as other relevant aspects related to humanization of the hematopoietic system of mice, i.e. any potential effects of human plasma on the clearance of human cells and the transfusion of different donor blood product.

The manuscript is clear, well written and the conclusions are supported by the data. Howerver, in order to make it suitable for publication I strongly encourage the authors to make some changes, more specifically:

In the headings of the result section I suggest to avoid the use of general terms such as "effect" and "roll" and already focus the attention on the main foundings that will be discussed in the paragraph.

All the headings were changed as suggested.

The figures must be ameliorated:

- most of them are kinda blurry and apparently submitted at low resolution (for example Fig. 1 and Fig. 3) please provide better quality images.

The figures have been submitted according to the very specific requirements of the journal. We believe high resolution images are available for download and will insure the final figures are of high quality.

verify their readibility; for example, Fig. 4 and Fig. 7 are organized in a confuse way and it is not easy to understand when the various panels start and finish.

For Fig. 4, we increased the spacing between sections to make the divisions more visually obvious. Instead of additional changes to the figure that we feel would result in either a lot of white space or a clutter of lines, we amended the legend (line 343) to clearly state the cytokines included in each section.

For Fig. 7, we reworked the figure to more clearly define the sections. 

Fig. 2 must be shifted to supporting informations.

Given that PLOS One is published electronically, we have left the figure within the manuscript so that readers do not need to download a separate file to view data that conveys important observations on the rapid hemolysis that occurs in wild type mice.

Responses to Reviewer 2

The paper by "blessinger et al, entitled: immunodeficient mice are better for modelling the transfusion of human blood compoments than wild type mice” aims to study the efficiency of blood cell transfusion in immunodeficient mice. The paper discusses different aspects impacting transfusion efficiency: the dose of blood cells to be transfused, the influence of age and weight of the mice ...

the experiments are well conducted, but questions remain unanswered and papers have already addressed these points in the literature, so this study lacks of novelty.

We appreciate the view that the finding that NSG mice are better hosts than immune competent mice may not be surprising to some, as this was our hypothesis when the work was initiated. We agree that for hematopoietic stem cell transplants it has been well documented that the series of mutations found in the NSG mice have combined to incrementally improve mouse models of hematopoiesis. However, in a search of the literature as well as in discussions with colleagues in the field of transfusion, there has been little formal demonstration that immunodeficient mice offer any advantages over mice with wild-type immune systems. This work was born from this controversy and we believe Fig. 1 of the manuscript answers the question definitively. If the reviewer is aware of any notable missing papers from our literature review in the introduction, we will be glad to incorporate them into the manuscript.

The questions that should be addressed are the following:

what happens in the first 5 minutes after transfusion? and that is the half-life of each blood cell after transfusion, a follow-up over several days is required.

New data was added regarding the frequencies of human cells in the first 5 minutes in Fig. 9C. These data are presented in lines 492-498 and show that peak frequency measurements can be obtained around 2 minutes after transfusion. 

We also analyzed experiments performed for the relationship between the weight of the mice and the levels of chimerism at time of first measurement (5-10 minutes). The results shown in the new Fig. 10 (lines 499-567) show a linear relationship in red cell transfused mice and the weight of the recipients as one would predict. However, mice transfused with leukocyte-enriched whole blood fail to show the same effect. These results provide more evidence that in the first minutes after transfusion, the circulation of human cells is rapidly affected in whole blood recipients with the levels of chimerism not strictly reflecting the blood volume of the mice. This affects all blood cell types, not just leukocytes. An interesting exception to the findings with LR-pRBC transfusions was the transfusion of a low dose of cells (Fig. 10I). As the previous manuscript stated, the low dose of cells showed evidence of a rapid clearance of transfused cells and the new data show that this coincides with a tendency of smaller mice having lower chimerism levels. 

Furthermore, we performed additional experiments to examine the clearance and tissue sequestration of LR-pRBC within the first 5 minutes of transfusion (new Fig. 12). These data show the liver and lungs as sites for human RBC sequestration. These data are presented in the final subsection of the Results (lines 644-664).

Half-life calculations were made and are presented in a new Table 2 (lines 636-643) and Supplementary Figure 2 (S2 Fig). 

The longevity of the transfused cells was further explored in an additional experiment. As shown in the new Fig. 11D-F, human red cell frequencies were at background levels at days 2 and 3 after transfusion. These results are in line with the previously shown data (now Fig. 11A-C). These data presented in lines 628-635.

are the transfused elements functional in these mice?

We previously showed that transfused human platelets can prevent vascular endothelial cell permeability resulting from VEGF administration. We added this to the Discussion (line 747-748). Beyond demonstration of the circulation of the human red cells, as well as many previous reports of human erythropoiesis in NSG mice, we are not able to demonstrate any additional functional attributes of the transfused erythrocytes in this model.

in addition to transfusion yield expressed as a percentage, what is the approximate number of cells recirculated, this is an important point to address to support the importance of using immunodeficient mice, analyses with count beads used in flow cytometry may address this question. 

We added estimates in the final paragraph of the Results (lines 665-674). For one of the estimates we used the transfusion of a known number of EGFP+ mouse red cells to estimate blood volume and recovery of human cells. We found that the blood contains far more human cells, than found in the liver, lung or spleen. However, the estimates do vary and some of the transfused human cells are ‘unaccounted’ for. 

How is the rebound effect of red blood cells and platelets explained? What is happening? What mechanisms are involved?

We speculated on possible mechanisms in the 3rd paragraph of the discussion (starting on line 704). The observation by Francis et al (2019) that we referenced suggests transfused cells are also found to be sequestered in tissues of humans receiving autologous transfusions. This effect, likely amplified by leukocyte cell adhesion as a result of xenogeneic transfusion and/or DIC are hypothesized mechanisms. However, further study of this complex phenomenon is beyond the scope of this work. The rebound effect is also not a major problem for modeling transfusion – the main focus of this work, as leukocytes are generally removed from blood products.

the use of clodronate has been discussed in several papers and has been shown to allow better recirculation of the figurative elements of blood without impairing the functionality of the cells, the point should be addressed in the experiments.

We have discussed the role of macrophages in human cell clearance in the sixth paragraph of the Discussion (starting on line 749). Although effective in increasing the survival of human cells, we raise the issue that elimination of the cells responsible for removal of old and damaged cells from the circulation likely diminishes the utility of the model for testing the quality of blood products. Line 755: “Thus, a macrophage-depleted mouse model will likely not accurately measure the quality of human blood products as the cells most critical in the removal of aged or damaged cells have been severely depleted.”

---

## [Decision Letter · Decision Letter 1]

21 Jul 2020

Immunodeficient mice are better for modeling the transfusion of human blood components than wild-type mice

PONE-D-20-05120R1

Dear Dr. Muench,

We’re pleased to inform you that your manuscript has been judged scientifically suitable for publication and will be formally accepted for publication once it meets all outstanding technical requirements.

Kind regards,

Francesco Bertolini, MD, PhD

Academic Editor

PLOS ONE

Additional Editor Comments (optional):

Reviewers' comments:

Reviewer's Responses to Questions

**Comments to the Author**

1. If the authors have adequately addressed your comments raised in a previous round of review and you feel that this manuscript is now acceptable for publication, you may indicate that here to bypass the “Comments to the Author” section, enter your conflict of interest statement in the “Confidential to Editor” section, and submit your "Accept" recommendation.

Reviewer #1: All comments have been addressed

2. Is the manuscript technically sound, and do the data support the conclusions?

Reviewer #1: Yes

3. Has the statistical analysis been performed appropriately and rigorously? 

Reviewer #1: Yes

4. Have the authors made all data underlying the findings in their manuscript fully available?

Reviewer #1: Yes

5. Is the manuscript presented in an intelligible fashion and written in standard English?

Reviewer #1: Yes

6. Review Comments to the Author

Reviewer #1: The authors fully addressed all my concerns. In general, this is a well conceived and carefully performed work, publication is recommended.

7. PLOS authors have the option to publish the peer review history of their article (what does this mean?). If published, this will include your full peer review and any attached files.

Reviewer #1: No

---

## [Editor Report · Acceptance letter]

23 Jul 2020

PONE-D-20-05120R1 

Immunodeficient mice are better for modeling the transfusion of human blood components than wild-type mice 

Dear Dr. Muench:

I'm pleased to inform you that your manuscript has been deemed suitable for publication in PLOS ONE. Congratulations! Your manuscript is now with our production department. 

Kind regards, 

on behalf of

Dr. Francesco Bertolini 

Academic Editor

PLOS ONE